# *Leishmania* Ribosomal Protein (RP) paralogous genes compensate each other's expression maintaining protein native levels

Francisca S. Borges[1☯], José C. Quilles, Jr[1☯], Lucas B. Lorenzon[1], Caroline R. Espada[1], Felipe Freitas-Castro[1], Tânia P. A. Defina[1], Fabíola B. Holetz[2], Angela K. Cruz[1]*

**1** Department of Cell and Molecular Biology, Ribeirão Preto Medical School, FMRP/USP–University of São Paulo, São Paulo, Brazil, **2** Laboratory of Gene Expression Regulation, Carlos Chagas Institute, Oswaldo Cruz Foundation, Curitiba, Paraná, Brazil

☯ These authors contributed equally to this work.
* akcruz@fmrp.usp.br

**Data Availability Statement:** All relevant data are within the manuscript and its Supporting Information files.

## Abstract

In the protozoan parasite *Leishmania*, most genes encoding for ribosomal proteins (RPs) are present as two or more copies in the genome. However, their untranslated regions (UTRs) are predominantly divergent and might be associated with a distinct regulation of the expression of paralogous genes. Herein, we investigated the expression profiles of two RPs (S16 and L13a) encoded by duplicated genes in *Leishmania major*. The genes encoding for the S16 protein possess identical coding sequences (CDSs) and divergent UTRs, whereas the CDSs of L13a diverge by two amino acids and by their UTRs. Using CRISPR/Cas9 genome editing, we generated knockout (Δ) and endogenously tagged transfectants for each paralog of L13a and S16 genes. Combining tagged and Δ cell lines we found evidence of differential expression of both RPS16 and RPL13a isoforms throughout parasite development, with one isoform consistently more abundant than its respective copy. In addition, compensatory expression was observed for each paralog upon deletion of the corresponding isoform, suggesting functional conservation between these proteins. This differential expression pattern relates to post-translational processes, given compensation occurs at the level of the protein, with no alterations detected at transcript level. Ribosomal profiles for RPL13a indicate a standard behavior for these paralogues suggestive of interaction with heavy RNA-protein complexes, as already reported for other RPs in trypanosomatids. We identified paralog-specific bound to their 3'UTRs which may be influential in regulating paralog expression. In support, we identified conserved *cis*-elements within the 3'UTRs of RPS16 and RPL13a; *cis*-elements exclusive to the UTR of the more abundant paralog or to the less abundant ones were identified.

## Introduction

Ribosomal proteins (RPs) are essential components of the ribosome in all organisms. In eukaryotes, 47 and 33 different RPs are present in the large (RLS) and small ribosome (RSS)

**Funding:** This project was supported by Fundação de Amparo à Pesquisa do Estado de São Paulo, FAPESP (https://fapesp.br/en) (2018/14398-0, 2015/13618-8, grants to A.K.C.); Brazilian National Council for Scientific and Technological Development (https://www.gov.br/cnpq/pt-br), CNPq (305775/2013-8) to A.K.C. This study was also supported in part by the Coordenação de Aperfeiçoamento de Pessoal de Nível Superior − Brasil (CAPES, https://www.gov.br/capes/pt-br), Finance Code 001, A.K.C. During of this work, F.S. B. (Grant 2019/0527-6), J.C.Q.J. (Grant 2020/00088-9), L.B.L. (Grant 2016/00969-0) and C.R.E. (Grant 2020/00087-2) were supported by FAPESP fellowships. The funders had no role in study design, data collection and analysis, decision to publish, or preparation of the manuscript.

**Competing interests:** The authors have declared that no competing interests exist.

subunits, respectively [1]. In *Saccharomyces cerevisiae*, approximately 75% of genes encoding for RPs are duplicated; these copies are either exact or highly similar in sequences with current findings suggesting that such duplicated genes confer benefits by altering gene dosage or evolving diverged functions [2]. Duplicated gene sequences that are identical or display high conservation at the level of the DNA coding sequence (CDS) but possess divergent untranslated regions (*UTR*s) may undergo distinct and paralog-specific gene expression regulation [3] whereas, dissimilarities in the CDSs may coincide with functional differences amongst the paralogs [4]. However, information on the expression control of these genes in different organisms remains scarce.

In addition to their canonical functions, noncanonical activities of RPs have been reported in different cellular pathways and organisms [5–7], such as disturbing bud site selection [8] and cell development in yeast [9]. More recently, RPs were linked to altered signaling pathways and oncogenesis [10]. In colorectal cancer, for instance, inactivation or partial mutation of the isoform RPL22 promoted the upregulation of its homologous paralog RPL22L1, leading to increased drug resistance [11]. Furthermore, noncanonical functions have been attributed to RPL13a, a conserved eukaryotic RP that regulates the translation of specific mRNAs molecules in humans [12].

Ribosomes are highly conserved RNA-protein complexes responsible for translation [13]. Differences in the levels of the ribosomal subunits (RS) may directly affect translation rates in the cell, and indeed alterations to the expression of paralog RP genes has been shown to directly affects RS levels. In yeast, the two paralogues of both *RPL26* and *RPL33* genes share highly conserved CDSs ($> 98\%$), but in both cases only one of the two paralogs exerted a positive influence on RLS abundance [9]. Studies in other organisms have reported different functions for duplicated genes encoding ribosomal proteins, a demonstration of moonlight activity for RPs. Given that most RP genes are duplicated in the genome of *Leishmania*, we sought to investigate if two duplicated RP genes, both with divergent *UTR*s and one with two amino acid substitutions, play differing roles in *L. major*.

In trypanosomatids, almost all polymerase II protein coding genes are organized into polycistronic transcription units (PTUs) [14] that lack canonical promoters for the individual gene transcription [15]. Consequently, gene expression control primarily operates at the posttranscriptional level [16] via alterations to mRNA transport, stability, decay and rate of translation [17]. *Leishmania* parasites are adaptive organisms switching between mammalian and invertebrate hosts during their life cycle, and modulating gene expression for their adaptation and survival in the distinct hostile environments [18]. Co-transcriptionally, mRNAs are transspliced at the 5'*UTR* in a process coupled to the polyadenylation of the upstream gene [19]. Gene expression modulation in these parasites is strongly dependent on *cis* elements present within their 3'*UTR*, which are recognized by RNA binding proteins (RBPs) that direct the transcript to distinct fates [20]. Interestingly, the translation of RP genes in trypanosomatids was recently reported to be regulated by different proteins binding to their 5' or 3'*UTR*s [21]. Thus, divergences in the *UTR*s of duplicated genes may promote differential expression via changes in transcript stability and/or the control of protein translation. In *Leishmania*, multicopy genes are common [14], yet little is known about the factors involved in their differential expression. Herein, to shed light on this regulation, we examine the expression profiles of two duplicated RP genes with divergences only in the *UTR* (*RPS16* genes) or both, in the *UTR* and CDS (*RPL13a* genes). The two amino acid substitutions occurring between the L13a isoforms are a Phenylalanine (Phe10) and a Glycine (Gly16) from L13a_15 that are respectively replaced in the L13a_34 by Cysteine (Cys) and Serine (Ser), both potentially subjected to a diversity of post-translational modifications (PTMs).

## Materials and methods

### Sequence alignment

*L. braziliensis* RPL13a and RPS16 protein and gene sequences were obtained from TriTryp data base (https://tritrypdb.org/tritrypdb/app/). The respective sequences were used as inputs to perform multiple global alignment queries in Clustal Omega (https://www.ebi.ac.uk/Tools/msa/clustalo/) to a sequence comparison. The RP nomenclature used herein is based on the annotation of the *Leishmania* genome database (https://tritrypdb.org/tritrypdb/app/). If using the universal nomenclature system proposed by Ban et al (2014), in which homologous ribosomal proteins are assigned the same name, regardless of the species, L13a would be named uL13 and S16, uS9. For structural alignments, the predicted protein structures for both variants of L13 were obtained from AlphaFold database (https://alphafold.ebi.ac.uk/entry/Q4QFG2 for LMJF_15_0200 and https://alphafold.ebi.ac.uk/entry/Q4Q3B9 for LMJF_34_0860). The URLs corrresponding to mmCIF files were submitted to the online Pairwise Structure Alignment tool (https://www.rcsb.org/docs/tools/pairwise-structure-alignment) (https://alphafold.ebi.ac.uk/files/AF-Q4QFG2-F1-model_v4.cif for LMJF_15_0200 and https://alphafold.ebi.ac.uk/files/AF-Q4Q3B9-F1-model_v4.cif for LMJF_34_0860) and the default alignment method (jFATCAT-rigid) was applied to generate the superposition. Alignments were visualized using an integrated tool within the Pairwise Structure Alignment tool.

### Parasite culture, differentiation, and transfection

Procyclic promastigotes of *Leishmania major* strain LV39 (MRHO/SU/59/P) were cultivated in M199 medium (Sigma Aldrich) supplemented with 10% heat-inactivated fetal bovine serum. To obtain metacyclic promastigotes, procyclic forms were cultivated for 5 days in M199 and metacyclics were enriched from stationary phase cultures by Ficoll® gradient [22]. Transfections performed as described previously [23]. Briefly, $1x10^7$ promastigotes were resuspended in 100 μL Tb-BSF buffer and added to 60 μL of DNA generated by PCR (for Δ or tagging). All this volume was transferred to an electrolytic cuvette and transfection was performed using the X-001 program of Amaxa Nucleofector instrument (LONZA). Cultures were maintained in M199 at 26°C before selection with the appropriate drugs (16 μg/mL hygromycin B; 20 μg/mL blasticidin; 16 μg/ mL puromycin) on solid M199 media. After 15–20 days, individual colonies were collected and transferred to liquid M199 medium containing the respective drug of selection and homozygosity was confirmed by DNA extraction and conventional PCR and sequencing. Using specific primers (ST.3), the tagged regions were amplified by conventional PCR, cloned into a pCR4-TOPO plasmid and sequenced by the Sanger method with M13 primers, confirming the correct insertion of 3 copies of the *myc* tag (S6 Fig)

### Immunofluorescence

RP subcellular localization was analyzed by immunofluorescence: a total of $1.5x10^6$ cells were centrifuged at 1,400 x g for 5 minutes at RT, followed by washing with 500 μl of PBS. Cells were fixed for 10 minutes at RT in 500 μL of 3% paraformaldehyde in PBS, pelleted and washed once with 500 μL PBS. The pellet was resuspended in 100 μL of 0.1% glycine in PBS and 30 μL of the total cells were added to poly-lysine slides and left to adhere for 30 minutes. Then, fixed cells were permeabilized with 30 μl of 0.2% Triton X-100 in PBS solution for 5 minutes at RT, then washed 5x in 1x PBS. Blocking was performed for 30 min with 5% skimmed milk powder dissolved in TBS-T. α-*myc* (Sigma C3956) primary antibody was diluted in blocking solution (1:4000) ratio and incubated for two hours. After incubation, five washes were performed with 1x PBS then secondary antibodies conjugated with Alexa Fluor

488 (Invitrogen A11001) were 1:500 diluted in PBS and incubated for 30 minutes at RT protected from the light. Nuclei and kinetoplast staining were performed using HOESCHT (Invitrogen H3570) at 2 μg·mL$^{-1}$ for 15 min. Images were acquired on a Leica DMI6000B fluorescence microscope at 60x magnification and processed using the Fiji Image software J (https://imagej.net/Fiji/Downloads) [24].

## Scanning Electron Microscopy (SEM)

$10^7$ parasites were fixed for 2 hours in 3% paraformaldehyde and 2% glutaraldehyde in PBS supplemented with 0.9 mM CaCl$_2$ and 0.5 mM MgCl$_2$ at RT. The parasites were post-fixed in 2% OsO$_4$ for 2 h and incubated with a thiocarbohydrazide (TCH) solution for 10 minutes, followed by ethanol and acetone dehydration. Then, parasite cells were mounted on a support and subjected to gold-coated metal plating. Cells were analyzed using an electron microscope scan (JEOL-JSM-5200). Images were captured in the Electron Microscopy Multiuser Laboratory (Department of Cellular and Molecular Biology, Ribeirão Preto Medical School, USP).

## Western blotting

Parasites were pelleted (1,400 x g for 5 minutes at RT), washed once with 500 μL of cold-PBS supplemented with protease inhibitors (Roche), resuspended in 10 μL of extraction buffer[54] and boiled for 10 min. Total protein was quantified in Nanodrop One spectrophotometer (Thermo Scientific) by measuring the absorbance at 280 nm. For every protein sample, sample buffer was added [23] and boiled for 3 minutes. 30 μg of total protein for each sample was fractionated in a 12% polyacrylamide gel. Proteins were transferred to nitrocellulose membranes (GE Healthcare Life Sciences: 10600003) and blocked for 1 h with TBS-T buffer (Tris-saline-Tween buffer: 10 mM Tris, 100 mM NaCl, pH 7.6, Tween20 0.1%) containing 5% milk powder. Immune detection was performed with the appropriate primary and secondary antibodies, following the manufacturer's recommendations: α-*myc* (1:4000; Sigma C3956) and α-EF1α: (1:80000; Merck 05–235). Both primary antibodies were diluted in solution of TBS-T with 2.5% powdered milk and incubated for 1 h at RT, followed by incubation with secondary antibody under the same conditions. Membrane visualization was performed via chemiluminescence (ECL kit–GE Healthcare: RPN2232) and images were obtained on the ImageQuant LAS 4000 equipment (GE Healthcare). Band quantification was performed using ROI manager tool of Fiji ImageJ software to determine the band skew of each sample. Band skew is a parameter used to quantify the pixels of one image, and has no specific unit [25]. Increase in the band skew means higher pixel's intensity, here directly correlated to the protein amount. All band skews were compared between the samples.

## RNA extraction and transcripts quantification

Cells were pelleted (1,400 x g for 5 minutes at RT), lysed with TRIzol reagent (Invitrogen) and RNA extraction was performed using DirectZol RNA Miniprep kit (Zyme Research). Total RNA was treated with DNase Turbo (ThermoFisher Scientific) and RT-qPCR performed and analyzed as described by Freitas Castro and cols [26], using G6PDH as housekeeping genes for normalization.

## Pull-down assay

The regions corresponding to the 3'*UTR* of the *RPS16* and *RPL13a* genes were retrieved from TriTryp data base (https://tritrypdb.org/tritrypdb/app/). These sequences were cloned into pUC-56 plasmid between a T7 promoter and 4xS1m aptamer sequences, adapted from

Leppek's strategy [27]. Then, RNA was *in vitro* transcribed (MEGAscript T7 transcription kit–ThermoFisher AM1334). Thirty micrograms of purified RNA were immobilized on Streptavidin magnetic beads (NEB) at 4°C for 8h under orbital rotation. $10^8$ parasites were lysed on ice by physical pressure using a 19G needle with 1mL of SA-RNP-Lyse buffer (20mM Tris-HCl pH7.5, 150mM NaCl, 1.5mM $MgCl_2$, 2mM DTT, 2mM RNAse inhibitor, 1 protease inhibitor cocktail tablet, 1% Triton X-100). Biotinylated proteins were previously removed from the extract by incubating the lysed extract for 8h at 4°C with the streptavidin beads. Then, the supernatant was incubated with the bead-immobilized RNA sequences for 8h at 4°C. An empty plasmid with no sequence between T7 promoter and 4xS1m aptamer was used as RNA control to identify unspecific proteins. After that, the beads were washed three times with wash buffer (20mM Tris-HCl pH7.5, 300mM NaCl, 5mM $MgCl_2$, 2mM DTT, 2mM RNAse inhibitor, 1 protease inhibitor cocktail tablet), resuspended in 35 μL of Laemmli buffer [28] and boiled for 10 mins before application into 12% polyacrylamide gel. Samples were run at 110V until the samples reached the separation gel.

## Proteomic analysis, mass spectrometry, DataBase searching and criteria for protein identification

Gel bands containing the samples were sent to protein identification by mass spectrometry analysis (Proteomics Platform of the CHU de Québec Research Centre, Quebec, Canada). Three biological replicates were evaluated for each protein and for the control. Results were obtained and analyzed using the software Scaffold Protein. The list of identified proteins was filtered using a protein threshold of 99%, a peptide threshold of 95% and a minimum of 1 peptide identified for all the samples. Proteins interacting with the control RNA sequence were unconsidered and the results were based on the proteins specifically interacting with the 3'*UTR* sequences in triplicate. Detailed information on Mass Spectrometry, Database searching and criteria for protein identification have been described elsewhere [23].

## Starvation resistance assay

Nutritional stress was evaluated by incubating $10^6$ parasites per well in a 96 well plate in PBS for 4 h. After that, plate was centrifuged 1,400 x g for 5 minutes at RT, cells were resuspended in complete M199 with MTT (3-(4,5-DIMETHYL-2-THIAZOLYL)-2,5-DIPHENYL-2H-TETRAZOLIUM·BROMIDE) 1 mg·$mL^{-1}$ and incubated for 24 h at 26°C. After that, parasites were pelleted and 200 μL of DMSO was added to solubilize the tetrazolium crystals. Absorbance was measured at 560 nm considering the substrate conversion for cell non-starved as 100% of nutritional response. Experiments represent three biological replicates performed as quintuplicate technical replicates.

## Sucrose density gradient

Promastigotes extracts from *Leishmania major* under normal growth condition were fractionated on sucrose gradient [29]. Briefly, 5 x$10^8$ cells were previously incubated in cycloheximide 100 μg·$mL^{-1}$ for 10 min. Cells were kept on ice and washed once with TKM buffer (10 mM Tris, 10 mM KCl, 1 mM $MgCl_2$ and pH 7.5) supplemented with 100 μg·$mL^{-1}$ cycloheximide, 10 μg·$mL^{-1}$ heparin, 10 μg·$mL^{-1}$ E-64 (cysteine protease irreversible inhibitor–Sigma-Aldrich) and protease inhibitor cocktail (Roche). For polysome dissociation, cells were treated with puromycin 2mM. Cells were pelleted and 100 μL of lysis buffer (TKM supplemented with 10% surfactant NP-40 and 2 M sucrose) were lysed by 10 x up and down agitation with a P200 pipette, followed by centrifugation at 18,000 x g at 4°C for 10 min. The supernatant was added on the top of linear 10–50% sucrose density gradient prepared in the same buffer. The system

was centrifuged at 39,000 rpm at 4˚C for 2 h in a Beckman SW41 rotor. After centrifugation, the gradient fractions were collected using the ISCO gradient fractionation system with the same sensitivity in the instrument (1.0) and starting the gradient profile at the baseline of 30 cm on the ISCO paper for all gradient replicates, using 30 µL of each sample for blotting assays. For a semi-quantitative analysis, the scale with arbitrary values of A254 nm was inserted into the polysomal profile graph, since the ISCO instrument does not provide absolute absorbance values. To assign these values, we relied on the centimeter scale of the ISCO paper on which the gradients are plotted, considering the maximum absorbance as 1. The intensity values of the 80S and 40S peaks were determined considering the y axis values. Peaks height were estimated considering the values from their base to the top. In this way, the intensity of the 80S peak relative to the 40S peak can be determined.

### Statistical analysis

Statistical *t*-test (Mann–Whitney) and one-way ANOVA followed by Tukey's multiple comparison tests were performed using GraphPad Prism 8, considering as significant a *p*-value $<0.05$.

## Results

### RPs from paralog genes have different expression levels and undistinguishable subcellular distribution

We investigated the expression of two pairs of duplicated paralogous genes encoding for RPs in *L. major* LV39 parasites: RPS16 (Fig 1A) and RPL13a (Fig 1B). *RPS16* was studied as a model for paralogs with identical coding sequences and divergent *UTR*s (Fig 1C), whilst *RPL13a* we selected as a representative for paralogs with non-identical coding sequences and divergent *UTR*s (Fig 1D). Seven nucleotide substitutions were found when comparing the two copies of *RPL13a* genes; two of them led to a codon modification and to non-conserved amino acid substitutions (Fig 1D) however, such alterations lead to no detectable changes in their 3D structures (S1D Fig), suggesting the RPL13a paralogs may retain conserved functions. Next, we focused on the roles of the divergent 3'*UTR*s, which are known to be involved in the interaction with RNA binding proteins [30] and, therefore, in the control of gene expression by regulating or modulating the mRNAs transcriptional and translational rates, as well as their stability.

To investigate the fate of each of these RPs in *L. major*, we used CRISPR/Cas9 to fuse an endogenous epitope tag to the N-terminus of each protein as previously described [31]. Briefly, donor DNA containing 3x*myc* epitopes and a blasticidin resistance gene was used to repair the Cas9-induced double strand break at 5'-end of each one of the paralogs (Fig 2A). The insertion of the tag in each copy of the RPS16 paralogs was confirmed by PCR and sequencing, showing that each individual paralog, RPS16_80 or RPS16_90, was efficiently tagged in both alleles (S1 and S6 Figs). For the RPL13a isoforms, due to the similarity of their amplicon size, the PCR was carried out for both copies, *RPL13a_15* and *RPL13a_34*, allowing the confirmation of tag insertion in only one paralogue with the maintenance of unmodified paralogous copy (S1 Fig). These confirmed transfectants were subsequently used to check the individual expression level and subcellular localization of all four RP isoforms.

Log-phase procyclic promastigotes and metacyclic promastigotes, purified from cultures in stationary phase by a Ficoll gradient fractionation [22], were isolated from axenic cultures and their morphologies examined by scanning electron microscopy to confirm their lifecycle stages; metacyclic promastigotes have elongated flagella and reduced cell body size when compared to the procyclic form (Fig 2B).

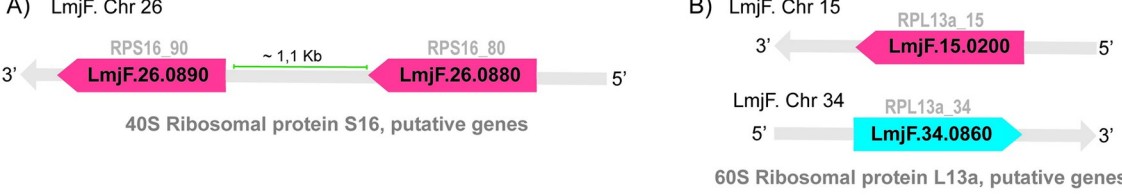

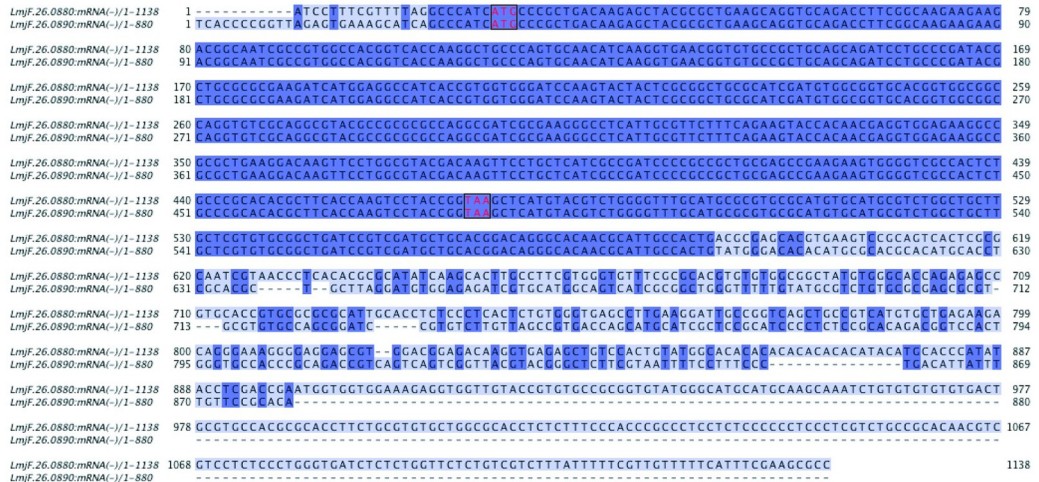

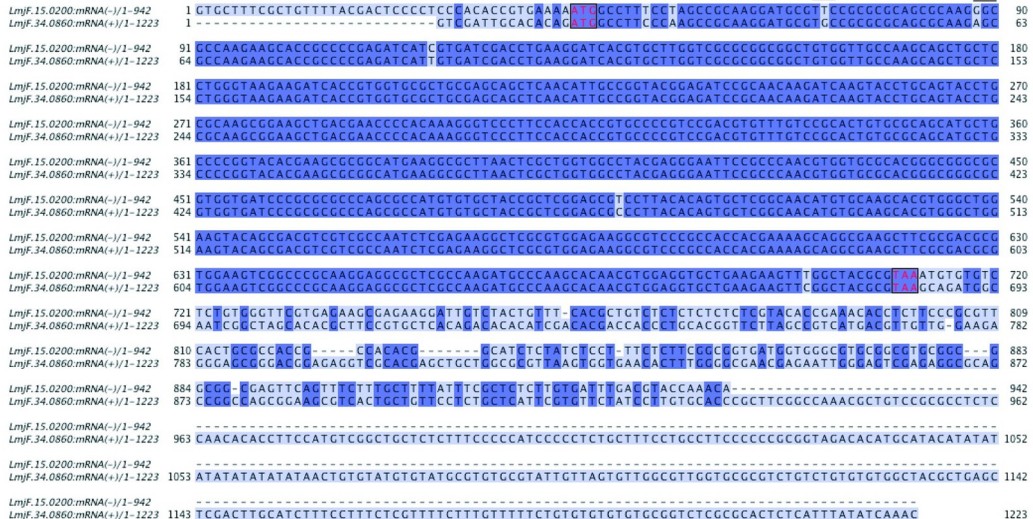

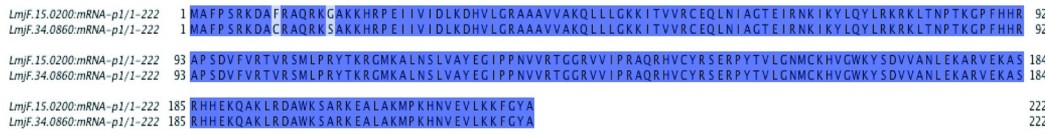

**Fig 1. Paralog genes and proteins alignments.** (A) 40S ribosomal protein S16 genes are found in tandem on the minus strand of chromosome 26 and (B) 60S ribosomal protein L13a genes are found on distinct chromosomes (15 and 34) and in opposite directions. (C) RPS16 and (D) RPL13a gene alignments, highlight the divergences in the UTR sequences. (E) RPL13a amino acid sequence alignment showing the two amino acid substitutions (*).

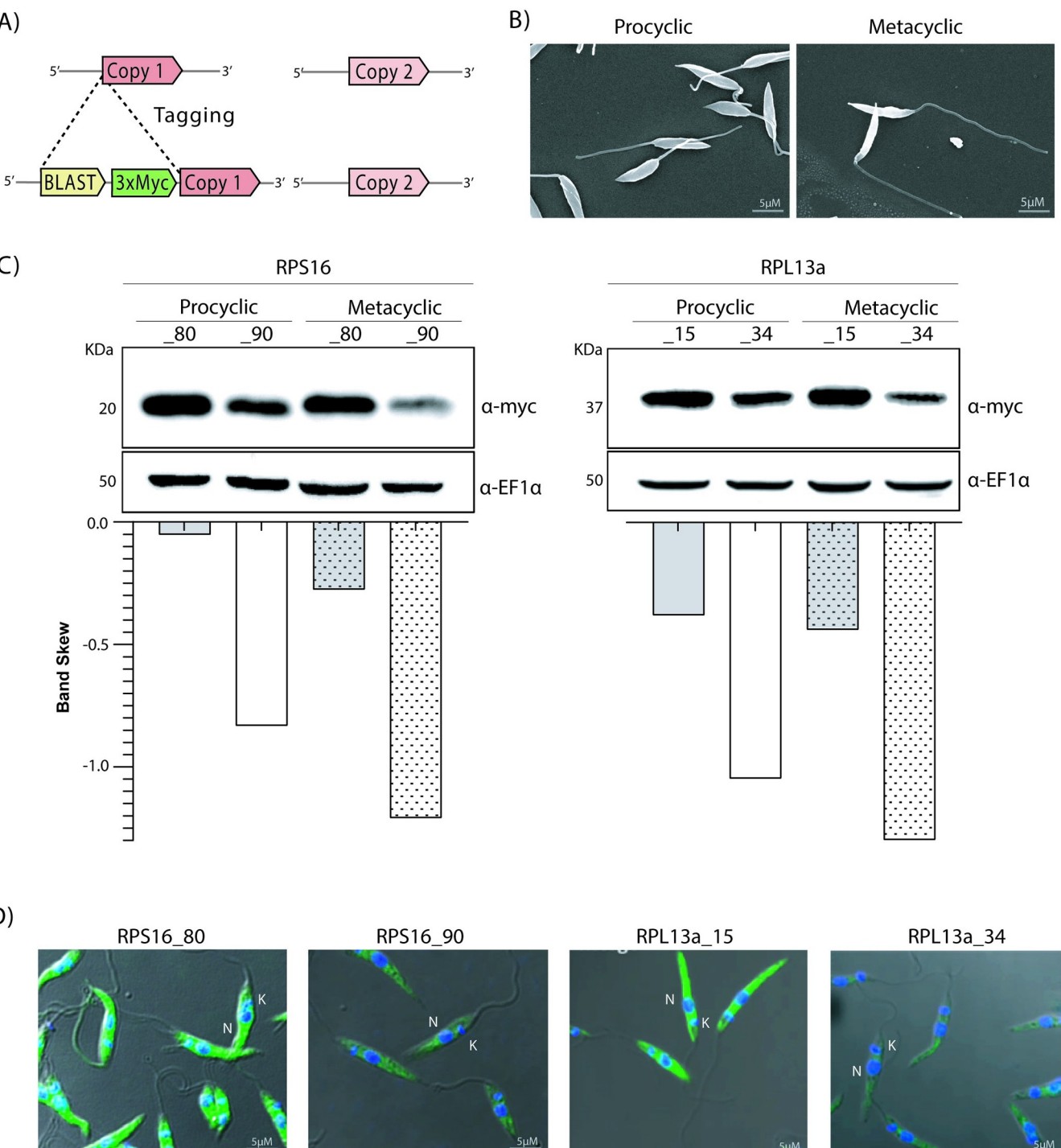

**Fig 2. Protein levels and subcellular localization of duplicated S16 and L13a ribosomal proteins.** (A) Strategy for 3x*myc* tag insertion in only one of the copies of the RP genes using CRISPR/Cas9. (B) Procyclic promastigotes were cultivated until the log phase from which metacyclic forms were purified by Ficoll gradient (see methods). Their morphological differences were confirmed by scanning electron microscopy. (C) Western blots probed with α-*myc* show that, for both RPs, the levels of protein for one paralog are consistently reduced in comparison to the other, and moreover, abundance is further reduced in metacyclic compared to procyclic promastigotes. (D) Immunofluorescence analysis confirms the cytoplasmic localization for all RP (green) proteins in procyclic promastigotes (nucleus [N] and kinetoplast [K] are indicated and stained in blue using Hoechst).

The relative protein abundance derived from each paralogous gene was evaluated by western blotting (Fig 2C) in both procyclic and metacyclic promastigotes. RPS16_80 and RPL13a_15 are consistently present in higher levels than their respective paralog in both lifecycle stages. Such differences in protein abundance were more prominent between the RPS16_80 and RPS16_90 isoforms in metacyclic promastigotes; RPS16_80 levels remain unchanged between procyclic and metacyclic stages but RPS16_90 was found to be lower in abundance when compared to RPS16_80; such difference between RPS16_80 e RPS16_90 levels is more striking in metacyclics.

The subcellular localization of RPS16 and RPL13a and their paralogs were investigated by indirect immunofluorescence. We detected *myc* signal for all RPs localizing (Fig 2D) exclusively to the cytoplasm, with no detectable co-localization with the nuclear or kinetoplastid Hoechst signal. Additionally, direct immunofluorescence performed for RPS16 using an α-RPS16 antibody raised in *L. major* LV39 confirmed the cytoplasmic localization in the *myc*-tagged RPS16 cell lines (S2 Fig). Importantly, the intensity of the immunofluorescence signal corroborated the western blotting results and confirmed the differences in the abundance of each RP isoform in procyclic promastigotes.

## Profile of proteins interacting with the UTRs of the studied RP transcripts

Since the 3'*UTR*s of both duplicated genes are not conserved and the expression levels of each paralog differed, we performed pull-down assays using the S1m *in vitro* system [27] to identify proteins potentially involved in such regulatory mechanisms. Briefly, the 3'*UTR* sequences were determined based on TriTrypDB annotations (Fig 3A), fused to the S1m aptamer and transcribed *in vitro*. After RNA immobilization on streptavidin-coated magnetic beads, the 3'*UTR* sequences were incubated with log-phase *L. major* LV39 protein extract. The bound proteins were isolated and identified by mass spectrometry (MS). Gene ontology (GO) analysis revealed that the proteins interacting with *RPL13a* and *RPS16* 3'*UTR*s are mostly involved in peptide and ribosome biogenesis, respectively (Fig 3B). Among these proteins, six were identified as binding to all four 3'*UTR* sequences (Fig 3C), and four of them are directly related to protein folding and ribosomal biogenesis and processing. We speculate that these four proteins common to all the examined *UTR*s are possibly core proteins that bind to mRNA of RP genes and might be involved in intra nuclear trafficking and nucleolar activities (Fig 3D).

## Increased response to starvation and RP levels of mutant parasites

Considering that ribosomal activity is affected under nutritional stress potentiating or uncovering non-canonical functions of ribosomal proteins, as shown for other organisms [5–8], we analyzed the resistance to starvation by comparing parental cells (pT007) to RP transfectants knocked out for each one of the paralogs. To this end, cells carrying a *myc* tag in one of the paralogous gene was knocked out for its corresponding counterpart (Fig 4A) with all knockouts (Δ) confirmed by PCR (S3 Fig). The ability to rapidly adapt and respond to starvation is critical for *Leishmania* differentiation and survival. Procyclic promastigotes were incubated for 4h in PBS, with recovery of 24h in fresh medium, and viability measured via colorimetric assays using MTT (Fig 4B). Curiously, an increment of the parasite resistance to the nutritional stress was observed for all the Δ parasites compared to parental cells, as indicated by the higher mitochondrial activity measured by MTT conversion (Fig 4C).

Additionally, the RP levels were evaluated by western blotting prior to and after 4 hours of starvation in PBS, as well as at 24hs post recovery in fresh medium. As shown (Fig 4D), upon starvation, a subtle decrease in the levels of both L13 and S16 proteins was observed, which were completely recovered to the unstarved levels after 24h in fresh medium. Interestingly,

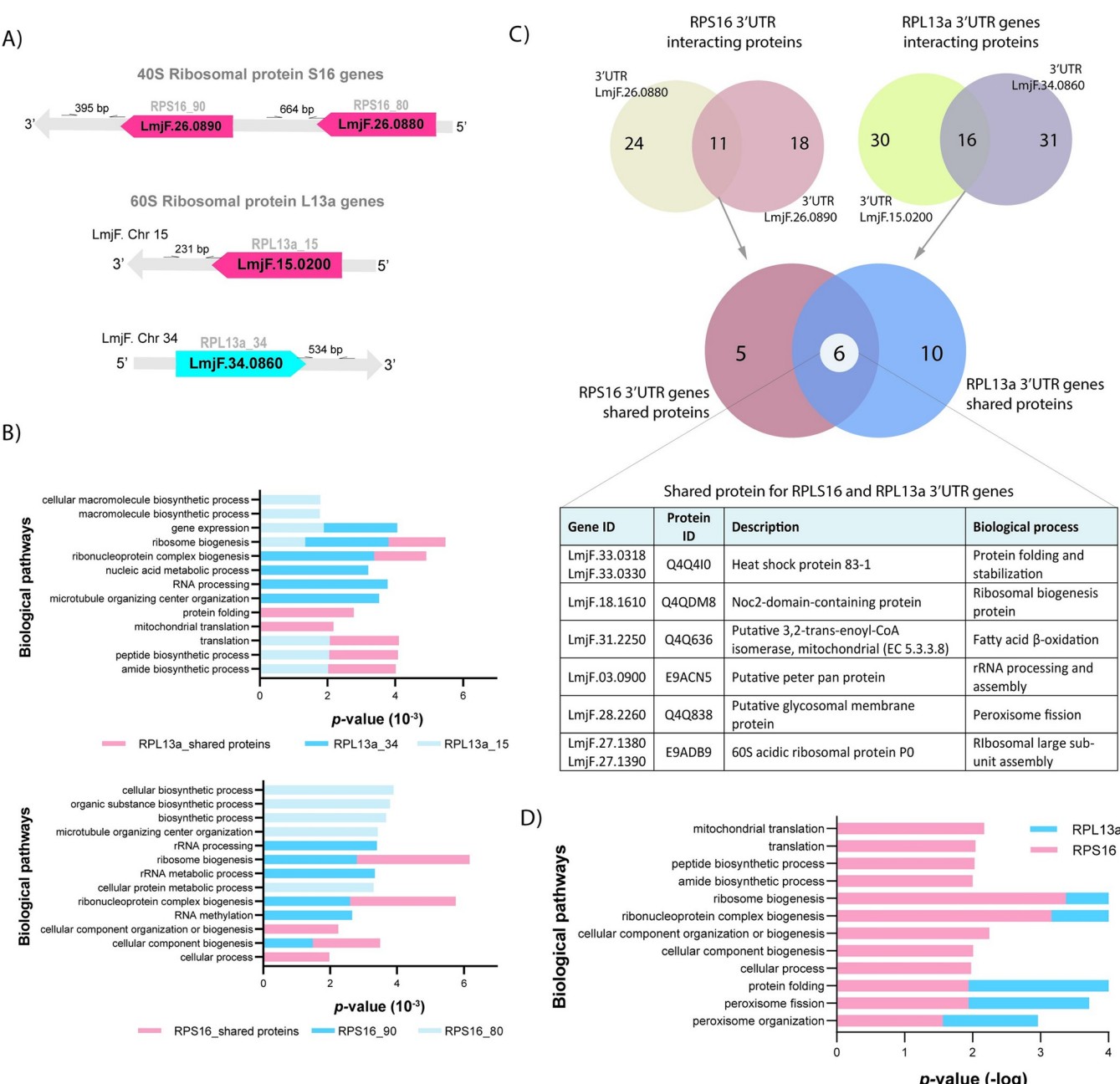

**Fig 3. Identification of proteins interacting with the 3'UTRs of the RPS16 and RPL13a paralogs mRNA.** (A) For the pulldown assays, fragments from the 3'UTRs of each gene were amplified and cloned into PUC-54-4xS1m; the position of the primers and lengths of the fragments are indicated. (B) Gene ontology analysis of proteins identified *in vitro* interacting with individual 3'UTR of each RPS16 and RPL13a paralog genes (shades of blue), and proteins interacting with both paralogs (red). (C) Specific and shared proteins for each duplicated gene were identified, with total of 24 and 18 for RPS16_80 and 90, respectively. For RPL13a paralogues, similar numbers of specific proteins were obtained: 30 and 31 proteins for RPL13a_15 and 34, respectively. (D) All 3'UTRs of ribosomal protein duplicated genes bound to proteins involved in ribosome pathways, when considered the five most relevant *p*-values (D). All the analyses were based on the results from three independent assays.

this trend was also observed for RPS16_90 and RLP13_34 which are consistently expressed at lower levels under physiological conditions (Fig 4D). Additionally, under stress conditions no marked changes in the subcellular distribution of these RPs were observed (Fig 4E).

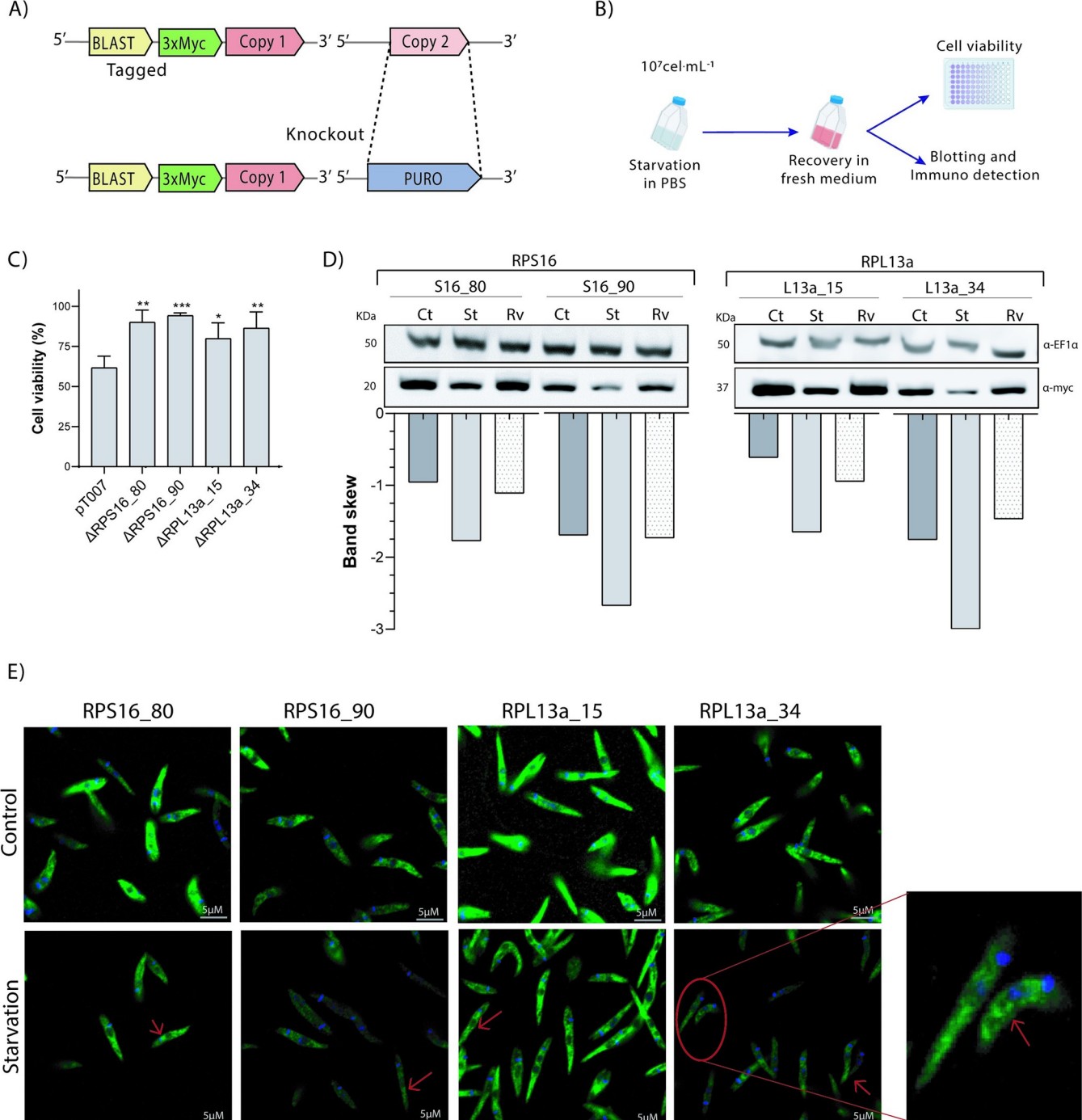

**Fig 4. Nutritional stress response and expression levels of RPs under starvation.** (A) After tagging one of the paralogues, the other one was deleted using CRISPR/Cas9. (B) Procyclic promastigotes in exponential growth were exposed to total starvation in PBS for 4h at 27°C, followed by viability assays, blotting and immunofluorescence analysis. (C) Cell viability was quantified by MTT assay at 24h post-starvation, to examine the recovery capacity of the mutant cells compared to parental line (pT007). (D) Expression levels of RPs were lower under starvation (St), particularly for RPS16_90 and RPL13a_34, with general recovery of parental levels after 24h of recovery in fresh medium (Rv) in comparison to the non-starved cells (control–Ct)–detection by western blotting with α-*myc*. (E) Immunofluorescence revealed no significant difference between the RPs distribution (green) under stress, but a subtle accumulation can be observed by increased signal in some regions of the cytoplasm (arrows) after 4h of total starvation (D).

## Knockout of RPL13a paralogs reduced the amount of 60S and 80S ribosomal subunits

Given that CDSs encoding the RPL13a proteins are non-identical with two non-conserved amino acids substitutions at the N-terminus (Fig 1E), we investigated the ribosome organization of these Δ cells under physiological condition. In that direction, we conducted polysomic profile analyses to investigate the status of the assembly of the ribosomal subunits, translation activity and biomolecule complexes [29,32]. Cell extracts of parental and Δ cells bearing only one of the *RPL13a* paralogs were fractionated by a sucrose gradient after treatment with cycloheximide or puromycin. For comparison purposes, we first analyzed the profile of polysomic distribution in parental and Δ cells in the presence of cycloheximide and observed a decreased peak for the 60S ribosome, as well as a subtle decrease in the 80S complex with a slight increase in the 40S subunit for the ΔRPL13a_15 and ΔRPL13a_34. As shown in Fig 5A, the intensity of the 80S peak for the ΔRPL13a_15 and ΔRPL13a_34 mutants relative to the 40S peak (80S/40S) is 2.2-fold and 1.6-fold, respectively. For the parental cell, 80S peak intensity relative to 40S peak (80S/40S) is 1.5-fold. When this 80S:40S ratio is compared between the three profiles, a 1.5- fold (2.2 or 2.3/1.5) reduction in the intensity of the peaks corresponding to the monosomal fraction of the both RPL13amutants is observed with respect to the parental strain. Additionally, polysomal peaks were significantly reduced compared to the parental cell line. These findings suggest that knockout of both RPL13a_15 and RPL13a_34 affects protein synthesis, even in the presence of the corresponding paralog. Next, we analyzed the distribution of each RPL13a paralog throughout the non-polysomic and polysomic fractions. No clear distinction in the distribution of a specific paralog in the absence of the other was observed, with both paralogous proteins being detected at similar levels at sucrose densities corresponding to monosomes and light or heavy polysomes (Fig 5B). Additionally, both RPL13a paralogs were detected mainly at the density of light polysome fractions when treated with puromycin, as if large and small subunits were assembled but not organized for translation (Fig 5C).

## Duplicated RPs present a paralog compensation that maintains protein levels

Next, we sought to check if different paralogs could compensate for the absence of each other in a compensatory mechanism to maintain the protein at parental levels. Protein levels were evaluated in both procyclic and metacyclic parental and knockout promastigotes by western blotting using a α-*myc* antibody. For all the studied proteins, when one of the genes was deleted, the tagged paralog protein levels increased compensating the lack of its counterpart (Fig 6A). This compensatory effect is more evident when the gene that is expressed at lower levels (*RPS16_90* and *RPL13a_34*) is kept and their highly expressed paralog deleted (Fig 2C). Additionally, to confirm that the presence of at least one of the paralogs was necessary for parasite viability, we proceeded to knock out both *RPS16* paralogs, but no double Δ clones could be recovered. Recovered clones had inserted two drug resistance genes from two rounds of transfections using CRISPR/Cas9 but all transfectants had retained at least one intact copy of the parental gene (S3 Fig).

In addition to protein abundance, gene expression was also evaluated at the transcript levels to evaluate at what level(s) the compensatory mechanism may occur. For that, total RNA from both ΔRP*L13a* procyclic promastigotes was extracted and transcript levels quantified by RT-qPCR utilizing specific primers for the divergent *UTR* regions of each L13a gene copy. Transcript levels in the Δ and parental cell lines were compared, and relative expression was normalized to the expression of the housekeeping gene Glucose-6-phosphate dehydrogenase (G6PDH). No significant differences in transcript levels were observed for either *RPL13a*

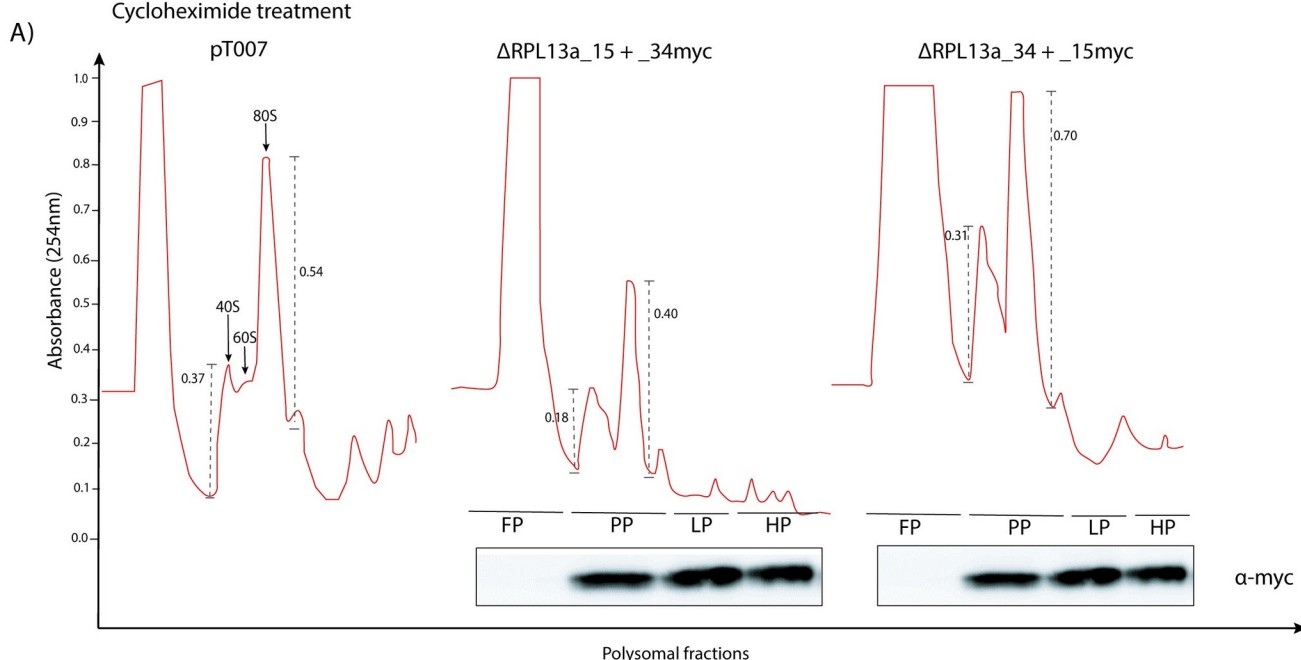

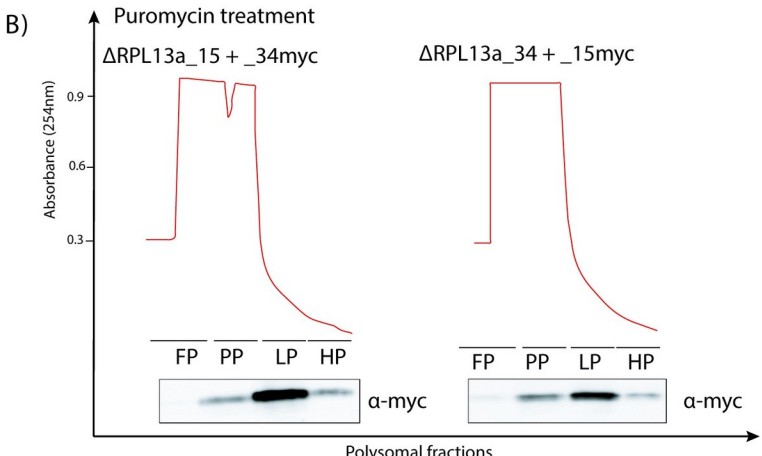

**Fig 5. Knockout of RPL13a paralogs decreases 60S, 80S and polysomes.** (A) RPL13a Δ mutants and the parental cell lines were subjected to sucrose gradient fractionation and polysomal profile determination, where FP: free polysomes fraction, PP: pre-polysomes, LP: light polysomes and HP: heavy polysomes. 254nm absorbance values were set arbitrarily and the relative intensity of the 80S and 40S peaks are shown in the dashed lines. Accumulation of both RPL13a isoforms in light polysomes fraction was determined by western blotting after cycloheximide treatment. (B) Additionally, both RPL13a isoforms were found in light polysome fraction (LP), even after ribosome dissociation by puromycin treatment. Experiments were performed in biological duplicate and the polysome profiles correspond to the average of each sample.

paralogs (Fig 6B). This result indicates that RPL13a compensatory mechanism involves either alterations to the translation rate or protein stability. However, it is unlikely that it is related to increases in transcript levels and/or stability.

## Discussion

The relevance of post-transcriptional gene expression regulation in *Leishmania* parasites [16] associated to the typically duplicated genes encoding for RPs raised the hypothesis that such proteins might be subjected to different expression regulation mechanisms or even possess

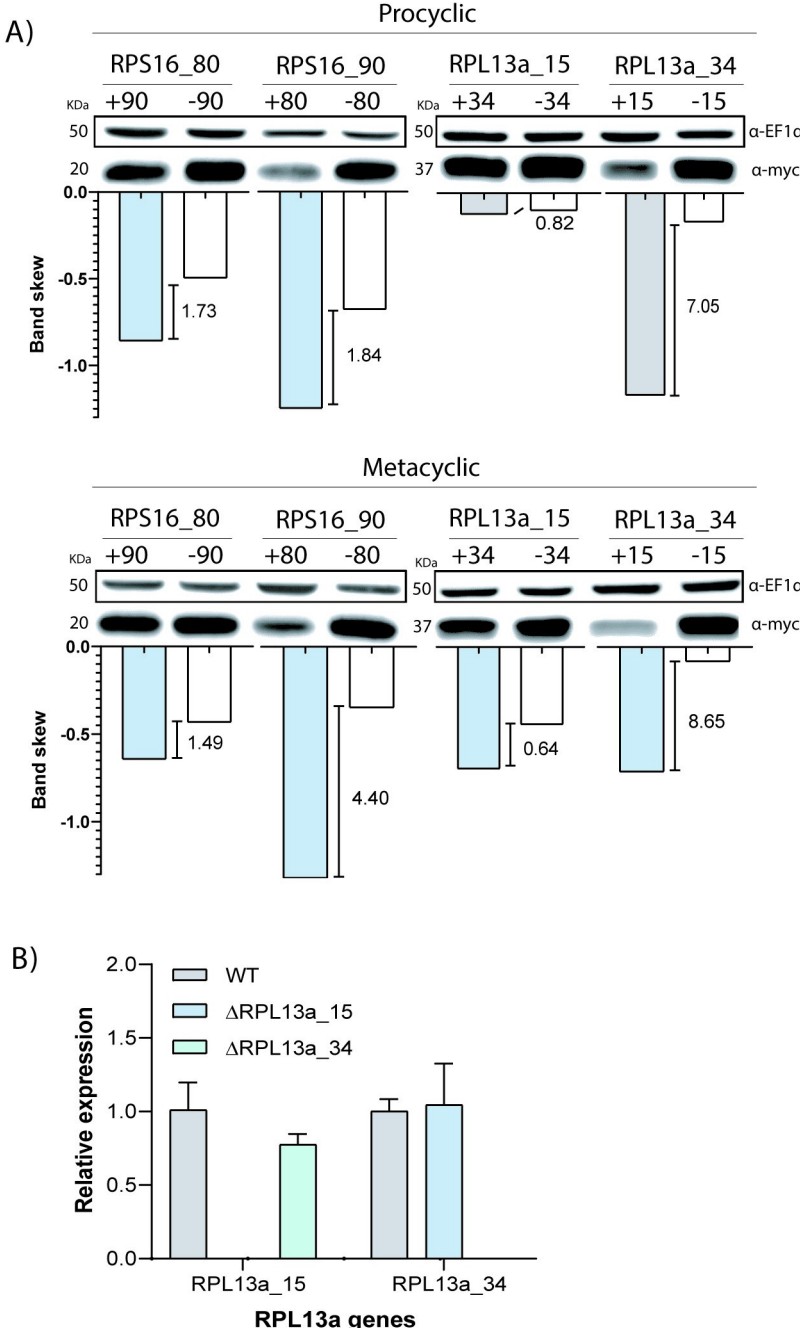

**Fig 6. Compensatory expression of RP paralogs in knockout cells.** (A) Expression levels in procyclic and metacyclic promastigotes were analyzed by western blotting using α-*myc*. The presence and absence (Δ) of the correspond RPS16 and RPL13a paralogues are indicated by + and -, respectively. Band intensity was quantified in ImageJ software and the results were plotted in the corresponding graph with the increment of band color indicated. (B) RT-qPCR was used to quantify the relative expression of RPL13a transcripts in the parental (WT) and Δ promastigotes. Expression of the gene of interest was normalized to the G6PDH housekeeping gene.

non-canonical functions. In our study, we observed a compensatory expression mechanism for two pairs of RP paralogous genes, *RPS16* and *RPL13a*, from *L. major*, which is linked to an increase in either the rate of translation or protein stability of each paralog, but without

measurable effects on the levels of the RPL13a transcripts. In humans, RPs are encoded by unique coding sequences, whereas 59 genes encoding for RPs are found to be duplicated in the *S. cerevisiae* genome [33]. Gene duplication was considered as a redundancy in the genome, since, in general, carrying two identical copies of a given gene was not considered to be advantageous [34]. However, for highly expressed transcripts, as RP genes–the most expressed in many cells [35]–such phenomenon can be favorable to supply the cell with high levels of RNA [34]. Gene duplication is currently understood as a recurrent and relevant process contributing to genome evolution [36]. The maintenance of duplicated RP genes with higher sequence similarity is remarkable when compared to other functional classes of duplicated genes in *S. cerevisiae*. The sequence conservation of RP paralogous copies may be attributed to their importance in maintaining protein folding or as surfaces for ribosome assembly and function [37]. Interestingly, these RPs typically have highly similar coding sequences but very different UTRs. In many of the investigated paralogs, the two copies were often expressed at different levels, after which the singular deletion of one copy of a pair led to different effects on fitness [37].

Herein, *Leishmania RPS16* and *RPL13a* genes were investigated at protein and transcript levels, with the aim to uncover evidence and add information on the regulatory processes of these genes. *RPS16* paralogs (*LmjF.26.0880* and *LmF.26.0890*) are localized *in tandem*, with identical CDSs and divergent *UTRs* (Fig 1A and 1C). *RPL13a* genes (*LmjF.15.0200* and *LmjF.34.0860*), on the other hand, are found on different chromosomes, with divergent *UTRs* and non-identical CDSs (Fig 1B and 1D). The divergent *UTRs* of the paralogs may suggest different expression regulation for these genes since untranslated regions contain sites for the binding of RBPs. 3'*UTRs* are strongly involved in the control of translational [30] and post-transcriptional [38] activities in eukaryotes and in prokaryotes, 3'*UTRs* are involved in gene expression regulation [39]. All this evidence strongly suggests that the control over the expression of these duplicated genes may be directly related to divergences within the 3'*UTRs*.

To investigate the relative abundance of RPS16 and RPL13a paralogs, all paralogs were tagged at N-terminal (Fig 2A). We consistently found higher levels of one of the paralogs across both procyclic and metacyclic forms, compared to their respective copy (Fig 2C). The abundance results observed for RPS16 isoforms corroborate those of RNA levels, as previously observed by RT-qPCR [26]. Due to the polycistronic nature of transcription in trypanosomatids, correlations between the levels of transcripts and proteins are often poor. Moreover, in human cells such correlations also appear poor, with only 33% of genes showing some correlation between protein and mRNA levels [40]; correlations are instead commonly observed for stage specific genes [41]. We were unable to evaluate the same parameters in amastigotes due to experimental limitations pertaining to the cultivation of *L. major* axenic amastigotes *in vitro*.

To search for any possibility of non-conventional functions for the RPs herein studied, we checked the nutritional stress response of mutant parasites, an essential process for the parasite adaptation and development in the in insect digestive tract [42], which also involves ribosomal activity. After 4h of starvation and 24h of recovery, MTT assays indicated higher cell viability for all the Δ parasite lines (Fig 4C). Increased mitochondrial activity could be a result of prolonged stress signaling, even after 24h of recovery. This elevated mitochondrial activity reveals a difference in resilience to nutritional stress in the knockouts compared to the parental cell line. Nevertheless, we cannot disregard the limitations of the MTT assay, as metabolic activity may be modified by external factors directly affecting the conversion of formazan in culture [43]. Although it cannot be ruled out at this point, no non-canonical roles for the RP isoforms investigated in this study were identified, but specific phenotypical alterations have been reported coinciding with the absence of one copy of a duplicated RP, *uL6A* or *uL6B*, in *S.*

*cerevisiae*, resulting in variations in general protein synthesis [44]. Furthermore, *RPL23AA* depletion promoted cell growth arrest with anomalies in *Arabidopsis thaliana*, even in the presence of its isoform *RPL23AB* [45].

During starvation, *Leishmania* parasites usually store their mRNAs and some ribosomal constituents within cytoplasmic granules to protect them from degradation [46]. The immunoblotting results suggest that no changes in RPs levels in the cell occur under starvation (Fig 4D), although a discrete, but detectable, clustering of these RPs suggests they may accumulate in some cytoplasmic aggregates (Fig 4E). In fact, loss of RPL13a isoforms caused a decrease in the 60S subunit, likely impacting upon the formation of the 80S subunit and subsequent polysomes. These data indicate a canonical behavior for the *Leishmania* RPL13a given a similar pattern of polysomic profile distribution was observed for *Trypanosoma cruzi* RPL26, which plays its canonical function [29]. However, in humans, RPL13a is dispensable for ribosomal functionality, but essential for mRNA methylation, a well-known non-canonical function for RPs [47]. Other non-canonical functions are also reported for human RPL13a, which plays different roles outside of the ribosome complex [48] including mRNA translation inhibition after RPL13a phosphorylation and its release from the ribosomal subunit [12] and as a constitute of the GAIT complex, a protein-RNA complex which drives the selective transcription control for a group of related genes [49].

Duplicated RPs might be associated with organism heterogeneity, and stage-specific RP isoforms were observed in *A. thaliana*, with higher RPS5A level found in rapidly dividing cells during the early embryonic development, compared with its isoform, RPS5B, which is preferentially expressed in differentiating cells [50]. In the same organism, RPL16b is another RP present in higher amounts in dividing cells, whilst RPL16A expression shows a tissue-specific association [51]. For *Leishmania* parasites, differentially expressed genes corroborate their agile adaptation to the different host environments during the life cycle. Interestingly, one of each RP paralog genes displays transcript and protein levels characteristically at higher levels than their corresponding paralog, and adjustment of the protein amounts to native levels consistently happens to compensate for the absence of the other paralog. We also show that this compensatory mechanism for both RPS16 and RPL13a occurs at the protein level when their paralogous copy was deleted (Fig 6A), similar to that observed for RPs from *S. cerevisiae* [44]. Such a mechanism does not involve the control of the transcript steady-state (Fig 6B). Thus, we speculate that this compensatory expression mechanism is linked to the canonical roles played by both isoforms and that there is a fine-tuning of protein abundance for each paralog. As previously mentioned, the lack of regulators at individual genes delegates control over gene expression in trypanosomatids largely to post-transcriptional processes, which includes mRNA steady-state, translation rate and protein stability or post-translational modifications [52]. Untranslated regions of mRNAs play a central role on these processes as they contain sites for the binding of regulatory proteins, the RBPs [53,54]. Herein, 11 and 16 proteins were identified to binding to the 3'*UTR*s of both *RPS16* and *RPL13a* isoforms, respectively (Fig 3C). After GO analysis, proteins involved in central aspects of translation, peptide and ribosome metabolism were identified in both 3'*UTR*s of *RPL13a* genes. For *RPS16*, the most relevant pathways were related to proteins involved in ribosomal biogenesis and metabolism (Fig 3B). Thus, these proteins identified by *in vitro* pulldown assay reinforce the hypothesis that the compensatory mechanism for the duplicated transcripts involves the translation machinery. Comparing the pool of proteins, we observed that only 6 of them are common to all four RP 3'*UTR* sequences analyzed (Fig 3C), and that they might be the core RBPome for RP transcripts. Interestingly, two proteins were detected to bind exclusively to the 3'*UTR*s of the isoforms present in lower abundance (*RPL13a_34* and *RPS16_90*) and eight proteins were only associated with the more abundant isoforms (*RPL13a_15* and *RPS16_80*). The last two sets of

proteins might be important for the specificities of control for the lower and higher RP expressors. In combination with these results, and with the aim to find shared and specific *cis*-elements which might act as binding sites for RBPs, we examined the 3'*UTR* sequences of the RP proteins identifying some conserved elements (S4 Fig) deserving of further experimental evaluation as putative functional binding sites implicated in the differential control of gene isoforms, as we have previously shown [55].

Despite a lack of evidence for non-canonical functions of *L. major* RPL13a in our report, such a hypothesis cannot be discarded given many prior reports have shown alternative roles for its orthologue in other organisms. Of note, the amino acid substitutions of Phenylalanine (Phe10) and Glycine (Gly16) in L13a_15 to Cysteine (Cys) and Serine (Ser) respectively, in the L13a_34 isoform, are putative targets of a large variety of post-translational modifications (PTMs). As for serine, various PTMs such as phosphorylation, sulfation, and various sugar chain modifications may occur. Additionally, the nucleophilicity and redox-sensitivity characteristic of cysteine residues results in a variety of PTMs. In contrast, Phe does not undergo PTMs, and Glycine is only subject to N-Myristoylation or N-acetylation. Despite the potential relevance to protein function of these substitutions [56], many PTMs are labile and dynamic, rendering them challenging to detect within a complex proteome [57]. Therefore, a technical limitation associated with the conditions used to evaluate PTMs and moonlight activities for each of the L13a isoforms may explain our results. In the end, we have provided insights into the regulation of gene expression for duplicated genes in *Leishmania* parasites, together supporting post-translational expression regulation. The conserved *cis*-elements within the 3'*UTR*s of these transcripts might be central to the compensatory mechanism observed for the paralogs at the protein level.

Overall, our results indicate that the RP duplicated genes studied compensate for the lack of the paralogous copy recovering the RP total levels of expression found in the wild type cells. Nevertheless, the non-identical L13 isoforms do not fully replace each other as we observed that despite the observed compensatory effect on the level of expression of the protein, the absence of one paralog has an impact on the formation of polysomes. This is an intriguing result to be further investigated.

## Supporting information

**S1 Fig. Strategy used for gene tagging and its confirmation for the ribosomal proteins under study.**
(DOCX)

**S2 Fig. Immunodetection of RPS16 using a specific rabbit antibody.**
(DOCX)

**S3 Fig. Confirmation of individual knockouts by conventional PCR for each of the duplicated genes.**
(DOCX)

**S4 Fig. Motifs found using the 3' UTR sequences for all RP transcripts.**
(DOCX)

**S5 Fig. Full western blotting membranes for all blottings presented in the article.**
(DOCX)

**S6 Fig. Sequencing results for all tag insertions.**
(DOCX)

**S1 Table. Proteins binding to the 3'UTRs of RPL13a duplicated genes.**
(DOCX)

**S2 Table. Proteins binding to the 3'UTRs of RPS16 duplicated genes.**
(DOCX)

**S3 Table. Primers used for all transfectionsRaw Data–MTT and RTqPCR values.**
(DOCX)

**S1 Raw data.**
(XLSX)

## Acknowledgments

We would like to thank Lissur A. Orsine for the computational analysis, Viviane Ambrosio for the technical support in the laboratory, the technical assistance at the microscopy facility (Ribeirão Preto Medical School), the staff of the Proteomics Platform of the CHU de Québec Research Centre, Quebec, Canada and TriTrypDB.

## Author Contributions

**Conceptualization:** Lucas B. Lorenzon, Angela K. Cruz.

**Data curation:** Francisca S. Borges, José C. Quilles, Jr, Lucas B. Lorenzon, Caroline R. Espada, Tânia P. A. Defina, Fabíola B. Holetz.

**Formal analysis:** Francisca S. Borges, José C. Quilles, Jr, Lucas B. Lorenzon, Caroline R. Espada, Felipe Freitas-Castro, Tânia P. A. Defina, Fabíola B. Holetz.

**Funding acquisition:** Angela K. Cruz.

**Investigation:** Francisca S. Borges, José C. Quilles, Jr, Fabíola B. Holetz.

**Methodology:** Francisca S. Borges, Caroline R. Espada, Felipe Freitas-Castro, Fabíola B. Holetz.

**Project administration:** Angela K. Cruz.

**Supervision:** Angela K. Cruz.

**Writing – original draft:** Francisca S. Borges, José C. Quilles, Jr.

**Writing – review & editing:** Angela K. Cruz.

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
