## [Decision Letter · Decision Letter 0]

2 Oct 2023

PONE-D-23-29797Leishmania Ribosomal Protein (RP) paralogous genes compensate each other’s expression maintaining protein native levelsPLOS ONE

Dear Dr. Cruz,

Thank you for submitting your manuscript to PLOS ONE. After careful consideration, we feel that it has merit but does not fully meet PLOS ONE’s publication criteria as it currently stands. Therefore, we invite you to submit a revised version of the manuscript that addresses the points raised during the review process.

Your paper has been evaluated by two reviewers. Please revise your paper according to the reviewers' comments. Please also respond to the reviewers' comments in a point-by-point manner that is easy for the reviewers to understand.

We look forward to receiving your revised manuscript.

Kind regards,

Keisuke Hitachi

Academic Editor

PLOS ONE

Reviewers' comments:

Reviewer's Responses to Questions

**Comments to the Author**

1. Is the manuscript technically sound, and do the data support the conclusions?

Reviewer #1: Yes

Reviewer #2: Partly

2. Has the statistical analysis been performed appropriately and rigorously? 

Reviewer #1: Yes

Reviewer #2: Yes

3. Have the authors made all data underlying the findings in their manuscript fully available?

Reviewer #1: Yes

Reviewer #2: Yes

4. Is the manuscript presented in an intelligible fashion and written in standard English?

Reviewer #1: Yes

Reviewer #2: No

5. Review Comments to the Author

Reviewer #1: The manuscript entitled "Leishmania Ribosomal Protein (RP) paralogous genes compensate each other’s expression maintaining protein native levels", is a very compact and elegant work that sheds light on the mechanisms of gene expression in trypanomatids, particularly in Leishmania. However, the manuscript can be improved in its presentation.

Some suggestions are:

1. Although the authors state that the study focuses on the role of different UTRs in regulation (lines 255-259), rather than function, since approximations are made to the possible non-canonical functions of the ribosomal genes under study (nutritional stress assays), it is suggested:

- Point out the a.a. that change between copies of RPL13a in the Introduction.

- To perform in silico models of the three-dimensional structure of RPL13a in order to discard possible structural changes affecting the protein function.

2. Avoid repeating the information provided to the reader in the Introduction in the Discussion.

3. The results concerning the Cis elements found in the UTRs are very important and, in fact, are included in the abstract. therefore, I suggest including them in Materials and methods and results and including the figure in the body of the manuscript.

4. Lines 328 to 331 correspond more to the discussion than to the results.

5. Clarify the stage of the parasites in Figure 3C.

6. RT-qPCR results of expression of RPS16 80 and 90 paralogs in mutant vs. wild-type parasites are not shown.

7. As for the discussion, I suggest changing the order. I mean, after presenting the two possible reasons for the parasite to have the same gene with two equal UTRs (lines 465 to 468), continue with the regulation issues, then the functionality, to finish with the compensatory effect.

8. Regardless of the stage, procyclic or metacyclic promastigotes, there is one copy of both genes that is expressed at the protein level more than the other. Could the length of the corresponding UTRs have something to do with it?

9.I wonder what the authors think about what would happen in the amastigote stage in relation to their research questions. I suggest including in the discussion.

10. It would be very interesting for the authors to discuss the possible effect of mutations at the level of promastigote replication, metacyclogenesis, infective capacity and amastigote replication.

11. Revise figures and supplementary material carefully as there are changes in gene or protein names, double legends, etc.

12. Include the database to which readers can refer to look up the raw data from the study, especially the pull-down findings.

Reviewer #2: Many ribosomal protein (RP) genes are duplicated in Leishmania. Dr Angela Cruz and coworkers have investigated the expression of the paralogous pairs of RP RPS16 and RPL13a during the transition from normal growth to starvation conditions. They report that the deletion of one of the two paralogous genes leads to increased protein expression from the other, though the corresponding change in mRNA levels is not observed. This agrees with the current model that most regulation of gene expression is post-transcriptional in Leishmania. Interestingly, microscopic experiments indicate that cytoplasmic r-proteins are distributed in several foci, which do not change during the transition from active growth to starvation. Finally, they analyze the ribosome organization through polysome gradient experiments. A peak sedimenting a little faster than the 80S is interpreted to represent a non-ribosomal RP-mRNA complex. The problem is interesting, but several aspects of the work need further investigation prior to publication.

Major issues

• Please mention the names of the RPs in the universal nomenclature (http://dx.doi.org/10.1016/j.sbi.2014.01.002) in order to facilitate comparison to results with other species.

• There are several problems with the polysome experiment (Figure 6)

o Show the OD trace from active growth and starvation conditions on comparative scales to allow a (semi-)quantitative comparison of the peaks.

o The peak moving slightly faster than the 80S particles runs similarly to the half-mers found in several other species. The authors should do a northern analysis of the fractions, at least in the 80S-disome range. If the authors’ interpretation is correct, no rRNA peak corresponding to the OD trace should be found between the 80S and the disome.

• The analysis of proteins binding to the 3’ORF should be continued with washes of incremental salt concentration to determine the hierarchy of affinities.

• Line 44: the statement is too general. The are many scries without such gene duplication, e.g., K. lactis and C albicans.

• The writing style should be tightened. The current version is quite conversational, at places it goes on extensive excursions for which there is little rigorous evidence (e.g., lines 598-603).

• Lines 197-198: samples did not enter the separation gel? How were the proteins then fractionated?

Minor issues

• The English language is generally good, but a review of the use of articles is suggested.

• Recommend more comparisons to yeast expression of paralogous RP.

6. PLOS authors have the option to publish the peer review history of their article (what does this mean?). If published, this will include your full peer review and any attached files.

Reviewer #1: **Yes: **Concepcion J Puerta

Reviewer #2: No

---

## [Author Response · Author response to Decision Letter 0]

16 Nov 2023

We thank the reviewers for the constructive criticism and comments that improved the manuscript and its contribution. 

Reviewer #1: 

The manuscript entitled "Leishmania Ribosomal Protein (RP) paralogous genes compensate each other’s expression maintaining protein native levels", is a very compact and elegant work that sheds light on the mechanisms of gene expression in trypanomatids, particularly in Leishmania. However, the manuscript can be improved in its presentation.

Some suggestions are:

1. Although the authors state that the study focuses on the role of different UTRs in regulation (lines 255-259), rather than function, since approximations are made to the possible non-canonical functions of the ribosomal genes under study (nutritional stress assays), it is suggested:

- Point out the a.a. that change between copies of RPL13a in the Introduction.

This information was added to the Introduction (lines 93-96) – “RPS16 proteins are identical while for L13a isoforms, a Phenylalanine (Phe10) and a Glycine (Gly16) present in L13a_15 are respectively replaced in the L13a_34 by Cysteine (Cys) and Serine (Ser), both subject to post-translational modifications (PTMs).”

- To perform in silico models of the three-dimensional structure of RPL13a in order to discard possible structural changes affecting the protein function.

As per reviewer #1 suggestion we performed in silico models using Alphafold and isoforms of RPL13a three dimensional structures suggest that protein structures are identical (SFig1D). 

2. Avoid repeating the information provided to the reader in the Introduction in the Discussion.

Where in the text we detected repeated information, we deleted or modified the text accordingly. These modifications are indicated in the version with track changes. 

3. The results concerning the Cis elements found in the UTRs are very important and, in fact, are included in the abstract. therefore, I suggest including them in Materials and methods and results and including the figure in the body of the manuscript.

We present these results as supplementary material because despite its putative relevance, we did not find statistically significant E-values (< 0.05), as indicated in the supplementary figure. In the revised manuscript we included the information in both methods and results sections. Nevertheless, given the significance of the result we maintained its representation in the supplementary material.

4. Lines 328 to 331 correspond more to the discussion than to the results.

The sentence below was removed from the results section. We did not transfer the paragraph to the discussion to avoid redundancy, since similar content was already included in the discussion.

5. Clarify the stage of the parasites in Figure 3C.

Parasite morphology information was added in the Figure subtitle.

6. RT-qPCR results of expression of RPS16 80 and 90 paralogs in mutant vs. wild-type parasites are not shown.

These assays were not performed since the CDSs are identical. We tested different combinations of primers but the few ones that could distinguish RPS16 80 from 90 did not amplify within an acceptable range.

7. As for the discussion, I suggest changing the order. I mean, after presenting the two possible reasons for the parasite to have the same gene with two equal UTRs (lines 465 to 468), continue with the regulation issues, then the functionality, to finish with the compensatory effect.

We accepted the reviewer’s suggestion, and the order was modified in the results and discussion. Thus, the items in the results were switched as blocks. For the sake of clarity, we did not highlight the block switches, but only the altered texts within these blocks.

Original order:

1. RPs from paralog genes have different expression levels and undistinguishable subcellular distribution

2. Duplicated RPs present a paralog compensation that maintains protein levels

3. Profile of proteins interacting with the UTRs of the studied RP transcripts

4. Increased response to starvation and RP levels of mutant parasites

5. RPL13a accumulates in non-polysomic fractions under starvation

New order:

1. RPs from paralog genes have different expression levels and undistinguishable subcellular distribution 

2. Profile of proteins interacting with the UTRs of the studied RP transcripts

3. Increased response to starvation and RP levels of mutant parasites

4. RPL13a accumulates in non-polysomic fractions under starvation

5. Duplicated RPs present a paralog compensation that maintains protein levels

8. Regardless of the stage, procyclic or metacyclic promastigotes, there is one copy of both genes that is expressed at the protein level more than the other. Could the length of the corresponding UTRs have something to do with it?

We found no correlation between 3’UTR length and levels of expression of corresponding proteins. In fact, L13a-34 and S16-90 are the ones presenting lower levels of expression compared to their paralogous copies, but S16-90 presents the shorter 3’UTR and L13a-34, the longest, as compared to the paralogous copy.

9.I wonder what the authors think about what would happen in the amastigote stage in relation to their research questions. I suggest including in the discussion.

Amastigotes, as procyclics, are host-specific replicative forms. Therefore, it is expected that their translational machineries are very active. Thus, we believe that the levels and the compensatory mechanism for these isoforms in amastigote will resemble those observed in procyclics, but at this point this is hypothetical. Notwithstanding relevant, we have experimental limitations for cultivation of L. major axenic amastigotes in vitro, being this the reason to restrict the study to the promastigote stages. 

10. It would be very interesting for the authors to discuss the possible effect of mutations at the level of promastigote replication, metacyclogenesis, infective capacity and amastigote replication.

Apart from the nutritional stress, for its potential correlation with ribosome and translation activity, we evaluated the polysome profile differences. We extracted metacyclics but did not conduct an analysis of metacyclogenesis neither infectivity or amastigote replication. 

11. Revise figures and supplementary material carefully as there are changes in gene or protein names, double legends, etc.

 We noticed that the description of figure 3A might not be clear to the reader, so we modified the description of the cartoon (lines 353-358). 

Another aspect that may have caused confusion is the Supplementary figure 5. We included SF5 as supplementary material, because all the membranes presented in the article were strips cropped from the entire membranes in all the figures mentioned in SF5. Thus, the aim of SF5 is to show that there are no hidden signals/bands, no manipulation of results obtained. 

12. Include the database to which readers can refer to look up the raw data from the study, especially the pull-down findings.

The file with all pull-down results was added to supplementary material. Raw data is in the process of deposit in the ProteomicXchange repository.

Reviewer #2:

Many ribosomal protein (RP) genes are duplicated in Leishmania. Dr Angela Cruz and coworkers have investigated the expression of the paralogous pairs of RP RPS16 and RPL13a during the transition from normal growth to starvation conditions. They report that the deletion of one of the two paralogous genes leads to increased protein expression from the other, though the corresponding change in mRNA levels is not observed. This agrees with the current model that most regulation of gene expression is post-transcriptional in Leishmania. Interestingly, microscopic experiments indicate that cytoplasmic r-proteins are distributed in several foci, which do not change during the transition from active growth to starvation. Finally, they analyze the ribosome organization through polysome gradient experiments. A peak sedimenting a little faster than the 80S is interpreted to represent a non-ribosomal RP-mRNA complex. The problem is interesting, but several aspects of the work need further investigation prior to publication.

Major issues

• Please mention the names of the RPs in the universal nomenclature (http://dx.doi.org/10.1016/j.sbi.2014.01.002) in order to facilitate comparison to results with other species.

We would like to justify not using the mentioned universal nomenclature on the original version. On the system proposed by Ban et al (2014), homologous ribosomal proteins are assigned the same name, regardless of species. They organized the nomenclature of ribosomal proteins to make it universal and easily recognizable irrespective of the origin. Nevertheless, Ban’s proposed nomenclature has not been broadly adopted by the community, and this was the reason for not employing it. More importantly, the nomenclature used in Leishmania is not the universal one, and we opted to use the nomenclature of the trypanosomatids annotated genomes, otherwise we would only increase entropy and confusion within the Leishmania community. Therefore, we added a brief explanation of the nomenclature issue, making the correlation of the Leishmania ribosomal protein identification with the corresponding Ban’s universal ones (L13a corresponds to uL13 and S16, to uS9). 

In the revised version of the manuscript, this information has been added to Materials and Methods (lines 108-112). 

• There are several problems with the polysome experiment (Figure 6)

o Show the OD trace from active growth and starvation conditions on comparative scales to allow a (semi-)quantitative comparison of the peaks.

As suggested by the reviewer, we have modified Figure 6A by inserting the scale with arbitrary values of A254 nm, since the ISCO instrument does not provide absolute OD values. To assign these values, we relied on the centimetre scale of the ISCO paper on which the gradients are plotted. Considering that the gradients were made based on the number of cells (5 x108), with the same sensitivity in the instrument (1.0) and starting the gradients at the baseline of 30 cm on the ISCO paper, we believe that our strategy for defining the scale for a semi-quantitative analysis is consistent with the reviewer's suggestion. In addition, we estimated the absorbance values to define the reduction of the 60S and 80S peaks relative to the profile of the parental strain. All these modifications are described in the Materials and Methods section and in the legend of Figure 6.

o The peak moving slightly faster than the 80S particles runs similarly to the half-mers found in several other species. The authors should do a northern analysis of the fractions, at least in the 80S-disome range. If the authors’ interpretation is correct, no rRNA peak corresponding to the OD trace should be found between the 80S and the disome.

We agree with the reviewer's observation that the peak moving slightly faster than 80S may resemble a halfmer. However, we do not state in the article that it is a disome or a halfmer. In fact, knockout of RPL13a paralogs caused a decrease in the 60S ribosomal subunit and consequently a reduction in the 80S monosome and polysomes, as observed for other ribosomal proteins in some organisms, suggesting a canonical role for RPL13a as a ribosomal protein. Although the polysomal profile of mutants from several different organisms knocked out for ribosomal proteins of the 60S subunit showed halfmers, we did not observe this similarity in the peaks related to polysomes in our gradients. Furthermore, when we looked at the disome peak in the gradient of the parental strain (where both copies of RPL13a are present), we observed the same distribution pattern for both mutants (Δ RPL13a_15 and Δ RPL13a_34). Regarding the definition of the area between 80S and disome, it should be noted that this technique does not have sufficient resolution to separate these fractions. Therefore, it would be difficult to obtain an rRNA-free fraction. In the article by Anish et al. (RNA (2012), 18:1968-1983), it is possible to verify the presence of rRNA in the range between 80S and disome, which supports our justification.

• The analysis of proteins binding to the 3’ORF should be continued with washes of incremental salt concentration to determine the hierarchy of affinities.

We thank the reviewer for raising this point, as it would be a thorough route to monitor the affinity force between proteins and UTRs. Such assay would be a step forward to improve our understanding of the putative interactions, along with RIP using tagged proteins, and other complementary assays for the analysis of RNA-protein interactions; they should improve precision to identify hierarchically protein binding profiles to distinct UTRs, but it is our understanding that these approaches should compose another study and are not on the scope of the present investigation. In the current study, and on the specific matter, our aim was to identify distinct protein profiles bound to the paralogous’ UTRs that could reinforce the hypothesis that these players, RBPs and UTRs, are in the core of the differential expression observed for the paralogous copies of the studied genes. Therefore, the affinity evaluation and complementary approaches as such abovementioned should be the object of a subsequent investigation. 

• Line 44: the statement is too general. The are many scries without such gene duplication, e.g., K. lactis and C albicans.

The paragraph has been modified to add precision (lines 43 to 50), specifically mentioning S. cerevisiae. In addition, a text on S. cerevisiae duplicated RP genes has been inserted in the first paragraph of Discussion.

• The writing style should be tightened. The current version is quite conversational, at places it goes on extensive excursions for which there is little rigorous evidence (e.g., lines 598-603).

We worked on the writing style aiming to improve precision and objectivity. The mentioned text at lines 598-603 have modified excluding fragments too speculative or misleading. 

• Lines 197-198: samples did not enter the separation gel? How were the proteins then fractionated?

For the mass spectrometry identification, the proteins were not fractionated in the resolving layer of the polyacrylamide gel, the electrophoresis is run to allow proteins to cross the stacking layer and line up only entering the resolving layer, to be collected as a unique band with all proteins. The in-gel proteins were sent to analysis in the Proteomics Platform of Laval University, Quebec, Canada, responsible for the sample preparation and peptide identification, as described. As previously mentioned, RAW Data is in the process of deposit in the ProteomeXchange repository.

Minor issues

• The English language is generally good, but a review of the use of articles is suggested.

The manuscript was sent to an expert for improving English language as suggested.

• Recommend more comparisons to yeast expression of paralogous RP.

As abovementioned, we modified the text in the introduction and discussion to highlight comparison with yeast RPs/duplicated genes.

---

## [Decision Letter · Decision Letter 1]

28 Nov 2023

PONE-D-23-29797R1Leishmania Ribosomal Protein (RP) paralogous genes compensate each other’s expression maintaining protein native levelsPLOS ONE

Dear Dr. Cruz,

Thank you for submitting your manuscript to PLOS ONE. After careful consideration, we feel that it has merit but does not fully meet PLOS ONE’s publication criteria as it currently stands. Therefore, we invite you to submit a revised version of the manuscript that addresses the points raised during the review process.

Please respond so that reviewer 2's concerns can be addressed.

We look forward to receiving your revised manuscript.

Kind regards,

Keisuke Hitachi

Academic Editor

PLOS ONE

Reviewers' comments:

Reviewer's Responses to Questions

**Comments to the Author**

1. If the authors have adequately addressed your comments raised in a previous round of review and you feel that this manuscript is now acceptable for publication, you may indicate that here to bypass the “Comments to the Author” section, enter your conflict of interest statement in the “Confidential to Editor” section, and submit your "Accept" recommendation.

Reviewer #1: All comments have been addressed

Reviewer #2: (No Response)

2. Is the manuscript technically sound, and do the data support the conclusions?

Reviewer #1: Yes

Reviewer #2: No

3. Has the statistical analysis been performed appropriately and rigorously? 

Reviewer #1: Yes

Reviewer #2: Yes

4. Have the authors made all data underlying the findings in their manuscript fully available?

Reviewer #1: Yes

Reviewer #2: Yes

5. Is the manuscript presented in an intelligible fashion and written in standard English?

Reviewer #1: Yes

Reviewer #2: Yes

6. Review Comments to the Author

Reviewer #1: (No Response)

Reviewer #2: The authors have addressed some of my issues, but the most serious criticism (Figure 5) stands. I cannot recommend publication until this is resolved.

1. Thank you for referencing the universal nomenclature. This facilitates the comparison to articles about other species.

2. I appreciate that the authors inserted scales for the A254 profiles of the sucrose gradients (Figure 5). However, my comment was not directed towards the comparison of strains, but rather the comparison of the profiles for M199 grown cultures (blue trace) with starved culture (red trace). The former peaks go off the scale and the patterns cannot be compared to the latter. Since the Materials and Methods indicate that 5x108 cells were harvested from both growing and starved cultures the very significant differences in signal intensity of the sucrose gradient A254 traces suggest that much more A254 material from the M199 cultures was loaded on the gradients as compared to the starved cultures. That is, the yield of A254 material was much greater from the M199 cultures than from the starved cultures. Why? Is the ribosome concentration decimated during starvation? A certain loss of ribosome density is expected but the extent indicated by the comparison of the blue and red traces is surprising. Is the lysis of the starved cultures inefficient? This issue should be further investigated by

a. tabulating the A254 yield of the cleared lysates and by quantitative northern analysis of rRNA in the lysates

b. rerunning the gradients with smaller amounts of material from the M199 cultures. It is essential to see how that trace changes during the starvation procedure. The pattern of the 40S-80S region does not look normal, even for the parental strain. Furthermore, the comparisons of peak heights in the revised manuscript is not understandable at all

c. running quantitative northern analysis on samples of all sucrose gradient fractions in order to identify peaks of the individual rRNA species.

3. The authors fail to address the issue of the peak running slightly ahead of the 80S. The peak does certainly not show the expected sedimentation rate of a disome peak (said on the basis of evaluating theoudsans of sucrose gradients over the years). I agree with the authors that the separation of the 80S and named peak is not complete, but running northern analysis of samples from each fraction would show if there is an rRNA peak that co-sediments with the named peak. If the authors are correct about a peak of L13 with mRNA, this should not be the case. The authors spend a lot of the text discussing potential extra-ribosomal functions of ribosomal proteins. Thus, it is essential to settle whether the experiments presented support this idea or not.

Other issues

1. Explain abbreviations such as E-64, MTT, TKM, and others throughout.

2. Line 44: human ribosomes have 47 60S ribosomal proteins.

3. Line 253: buffer lysis > lysis buffer.

4. Line 254: something missing after the end of the parenthesis.

5. Line 254: agitation in a pipette; add the type of pipette and how many times, which could affect lysis efficiency.

6. Figure 1: Define asterisks.

7. Line 298: are the insertions also confirmed by sequencing?

8. Quantify blots throughout.

9. Indicate the base value of the log base in Figure 3B.

10. Line 395: I suggest that the text be changed to “which were completely reversed”

11. Figure 4D and Figure 6: What are the units on the histogram scale?

12. Line 431: The 60S peak is already reduced relative to the 40S peak in the pT007 culture.

13. Line 458 and in Figure 5: Free polysome (FP)> polysome free (PF).

14. Figure 5C: Puromicyn> Puromycin.

15. Line 516: not up-to-date; see e.g. papers from Maria Barnes’ group

16. Line 572: The authors should mention the effect of extra-ribosomal proteins on p53 stability.

7. PLOS authors have the option to publish the peer review history of their article (what does this mean?). If published, this will include your full peer review and any attached files.

Reviewer #1: **Yes: **Concepción J Puerta

Reviewer #2: **Yes: **Lasse Lindahl

---

## [Author Response · Author response to Decision Letter 1]

14 Dec 2023

Ribeirão Preto, December 13, 2023

Alireza Badirzadeh

Academic Editor 

PLoS One

Dear Dr. Badirzadeh,

We here submit a second revised version of our manuscript entitled “Leishmania Ribosomal Protein (RP) paralogous genes compensate each other’s expression maintaining protein original levels” to PLoS One. In this new version, only one of the reviewers maintained criticisms that prevented him from recommending the article for publication. We answered to the points raised by the reviewer and modified the manuscript accordingly. 

We would like to bring to the editor attention that some of the critical points raised by the reviewer led us to modify the manuscript. Along with other changes detailed in the response to the reviewer, responding to his criticisms, we have chosen to delete delete the results related to polysome profiles under stress conditions, since given the pointed weaknesses of the data, a large set of assays would have to be conducted. We emphasize here that data related to stress and polysomic profile are not relevant to our study and do not affect our conclusions. As we detailed in our response to the reviewer, carrying out the experiments proposed by him would greatly contribute to elucidating the role of the L13 protein in the formation of the 60S subunit and in protein synthesis. However, this investigation does not fall within the scope of our work.

We clarify that Lissur Orsine, the bioinformatician in the laboratory, responsible for the Alphafold analysis of the protein’s structures, considered that her contribution to the manuscript did not support authorship and asked me to include her on the acknowledgments section only.

We hope that you find our manuscript, in its revised version, adequate and representing a robust contribution for publication in PLoS One.

We look forward to your response,

Yours faithfully,

Angela Kaysel Cruz

Professor

Dept of Cell and Molecular Biology

Email: akcruz@fmrp.usp.br

---

## [Decision Letter · Decision Letter 2]

20 Dec 2023

PONE-D-23-29797R2Leishmania Ribosomal Protein (RP) paralogous genes compensate each other’s expression maintaining protein native levelsPLOS ONE

Dear Dr. Cruz,

Thank you for submitting your manuscript to PLOS ONE. After careful consideration, we feel that it has merit but does not fully meet PLOS ONE’s publication criteria as it currently stands. Therefore, we invite you to submit a revised version of the manuscript that addresses the points raised during the review process.

Although this is after two revisions, just a few more minor revisions are required. Please respond appropriately before publishing.==============================

We look forward to receiving your revised manuscript.

Kind regards,

Keisuke Hitachi

Academic Editor

PLOS ONE

Journal Requirements:

Reviewers' comments:

Reviewer's Responses to Questions

**Comments to the Author**

1. If the authors have adequately addressed your comments raised in a previous round of review and you feel that this manuscript is now acceptable for publication, you may indicate that here to bypass the “Comments to the Author” section, enter your conflict of interest statement in the “Confidential to Editor” section, and submit your "Accept" recommendation.

Reviewer #2: (No Response)

2. Is the manuscript technically sound, and do the data support the conclusions?

Reviewer #2: Yes

3. Has the statistical analysis been performed appropriately and rigorously? 

Reviewer #2: Yes

4. Have the authors made all data underlying the findings in their manuscript fully available?

Reviewer #2: Yes

5. Is the manuscript presented in an intelligible fashion and written in standard English?

Reviewer #2: Yes

6. Review Comments to the Author

Reviewer #2: The authors have responded to my critique of Figure 5 by eliminating the results of the starvation experiment. I applaud his decision since it will take some time to complete this investigation and the starvation experiment is not essential for interpreting the expression of the L16a and L13 decision alleles. See however below.

The authors also addressed many of my comments under “Minor Points” but a few points need to be addressed.

1. Were the myc insertions sequenced?

2. Some blots, e.g. Figure 2c, should be quantified.

3. The estimates of the peak heights in Figure 5 are not done correctly. The baselines for the 40S and 80S peaks is not at the bottom of the “valleys” between peaks, because the different peaks overlap. The best way to compare is to overlay the A254 tracings, line up the corresponding peaks, and graphically adjust the 40S peaks to have equal height. The authors arrive at the correct conclusion, but the procedure in not correct. Here is an example:

(See attachment for figure)

4. Line 264-265: Absolute OD is not a proper term, since OD, most correctly called A (absorption), is defined as the log10 of the ratio between the light transmission in a control solution relative to the experimental solution.

7. PLOS authors have the option to publish the peer review history of their article (what does this mean?). If published, this will include your full peer review and any attached files.

Reviewer #2: **Yes: **Lasse Lindahl

---

## [Author Response · Author response to Decision Letter 2]

27 Dec 2023

Critique of the third submission – Reviewer 2

The authors have responded to my critique of Figure 5 by eliminating the results of the starvation experiment. I applaud his decision since it will take some time to complete this investigation and the starvation experiment is not essential for interpreting the expression of the L16a and L13 decision alleles. See however below.

The authors also addressed many of my comments under “Minor Points” but a few points need to be addressed.

1. Were the myc insertions sequences?

Myc insertions was performed by CRISPR/Cas9 system and confirmed by conventional PCR (please, see Supplementary material).

2. Some blots, e.g. Figure 2c, should be quantified.

As previously done for all blotting, these blots showing the expression of the both protein isoforms in procyclic and metacyclic morphologies were quantified, as requested, and the Figure 2 was updated.

3. The estimates of the peak heights in Figure 5 are not done correctly. The baselines for the 40S and 80S peaks is not at the bottom of the “valleys” between peaks, because the different peaks overlap. The best way to compare is to overlay the A254 tracings, line up the corresponding peaks, and graphically adjust the 40S peaks to have equal height. The authors arrive at the correct conclusion, but the procedure in not correct. Here is an example:

We apologize for our mistake during the peak quantification. Now, we present all peaks re-quantified according to the reviewer’s suggestion, as well as the updated version of the Figure 5.

4. Line 264-265: Absolute OD is not a proper term, since OD, most correctly called A (absorption), is defined as the log10 of the ratio between the light transmission in a control solution relative to the experimental solution.

As suggested, absolute OD term has been replaced by absorbance, once the ISCO equipment generates the peaks in the polysome profile based on the absorbance for each sample at 254nm. However, it does not provide the absolute absorbance values for the peaks, with the polysome profiles represented in a scale paper. So, all absorbance values for each peak were estimated based on the paper scale, from 0 to 1, same values commonly used to represent absorbance, according to Lambert-Beer low.

---

## [Decision Letter · Decision Letter 3]

2 Jan 2024

PONE-D-23-29797R3Leishmania Ribosomal Protein (RP) paralogous genes compensate each other’s expression maintaining protein native levelsPLOS ONE

Dear Dr. Cruz,

Thank you for submitting your manuscript to PLOS ONE. After careful consideration, we feel that it has merit but does not fully meet PLOS ONE’s publication criteria as it currently stands. Therefore, we invite you to submit a revised version of the manuscript that addresses the points raised during the review process.

We look forward to receiving your revised manuscript.

Kind regards,

Keisuke Hitachi

Academic Editor

PLOS ONE

Journal Requirements:

**Additional Editor Comments:**

Thank you for your efforts addressing the revisions. However, the reviewers' requirements have not yet been completely met. Please respond fully to the reviewers' comments before accepting the manuscript. I wish you a Happy New Year.

Reviewers' comments:

Reviewer's Responses to Questions

**Comments to the Author**

1. If the authors have adequately addressed your comments raised in a previous round of review and you feel that this manuscript is now acceptable for publication, you may indicate that here to bypass the “Comments to the Author” section, enter your conflict of interest statement in the “Confidential to Editor” section, and submit your "Accept" recommendation.

Reviewer #2: (No Response)

2. Is the manuscript technically sound, and do the data support the conclusions?

Reviewer #2: (No Response)

3. Has the statistical analysis been performed appropriately and rigorously? 

Reviewer #2: (No Response)

4. Have the authors made all data underlying the findings in their manuscript fully available?

Reviewer #2: (No Response)

5. Is the manuscript presented in an intelligible fashion and written in standard English?

Reviewer #2: (No Response)

6. Review Comments to the Author

Reviewer #2: Thank you for your responses to R2. However, I see two outstanding issues:

1. Please define the calculation and scale for "band skew. I brought this is in my comments on version R1.

2. The myc insertions have not been sequenced. I think this should be done.

7. PLOS authors have the option to publish the peer review history of their article (what does this mean?). If published, this will include your full peer review and any attached files.

Reviewer #2: **Yes: **Lasse Lindahl

---

## [Author Response · Author response to Decision Letter 3]

15 Jan 2024

Critique of PONE-D-23-29797_R3.

Reviewer #2: Thank you for your responses to R2. However, I see two outstanding issues:

1. Please define the calculation and scale for "band skew”. I brought this is in my comments on version R1.

We apologize for the laconic explanation given on R1. The band intensity on western blots was determined by ImageJ using “ROI manager” (ROI = Regions of Interest); the tool allows quantification of all bands, based on their size and intensity. Inside the ImageJ software, the array can be segmented into ROIs of identical size, where each ROI corresponds to one protein, certifying that the same area was considered for future comparisons between the samples. Thus, band skew shows how much the band intensity decreased comparing the RP expression in the presence or absence of its paralogue (Nat Protoc. 2016 August; 11(8): 1508–1530. doi:10.1038/nprot.2016.089).

2. The myc insertions have not been sequenced. I think this should be done.

We will ask reviewer #2 to reconsider this specific demand for the reasons below. We agree that sequencing knockouts, tagging or any genome editing is the most robust form of confirming the expected modification, without any further undesirable event, or off-target insertion. Nevertheless, I want to raise some points that might convince Reviewer #2 that, in this case specifically, we have robust evidence that both knockout and myc tags occurred as aimed. First aspect to consider is that genome editing in Leishmania has been first achieved in 1990, using the classical method of gene replacement that had been previously employed for Saccharomyces cerevisiae. We (Cruz and Beverley, 1990) pioneered the field of Leishmania genetic manipulation when we showed that “Following introduction of a construct containing dihydrofolate reductase–thymidylate synthase (dhfr-ts) flanking sequences fused to neomycin phosphotransferase, 45% of the colonies contained the planned homologous replacement; this frequency rose to nearly 100% in transfections using low amounts of DNA. Integrative transfection in Leishmania thus resembles that of Saccharomyces cerevisae in giving predominantly homologous events.” (http://dx.doi.org/10.1038/348171a0). This pioneer work demonstrated that the parasite has a highly efficient machinery of homologous recombination, which largely differs from most eukaryotes; insertion does not occur randomly and even when we exceeded in the amount of DNA to be transfected, what we observed was multiple insertions of the fragment in the homologous sequence/aimed site (same article above). The establishment of the CRISPR-Cas9 in these parasites (doi: 10.1098/rsos.170095) came to facilitate the previously labor-intensive and time-consuming approach, and represents a significant advance in the field. Even after 6 years of CRISPR/Cas9 use in Leishmania, we find no publications reporting off-target editing in Leishmania, and there are authors that report complete sequencing of the genome of transfected parasite, with no off-target events, this is probably partially due to the intrinsic genetic features of the organism. In addition to the abovementioned, we must consider that the results we report now compose a robust indication that myc insertion occurred in the expected genomic site.

The tag insertion in the RP genes studied in our work was also confirmed based on immune detection, as we used S16 antibody; the subcellular distribution of the tagged proteins using α-myc antibody was undistinguishable from the distribution of RPS16 protein detected by a specific α-S16 antibody (see supplementary material SF2). We have also used S16 polyclonal antibody in WB and the pattern was the same (not shown). The compensatory effect that maintains protein levels at similar levels comparing KO with wild type parasites was coherent, for all the paralogues (S16 and L13), showing that we are dealing with the protein we aimed to tag. For these reasons, we believe that the conventional PCR was an adequate tool in this study and confirmed the high efficiency of homologous recombination in Leishmania parasites, for both knockout and tagged parasites generation.

---

## [Decision Letter · Decision Letter 4]

22 Jan 2024

PONE-D-23-29797R4Leishmania Ribosomal Protein (RP) paralogous genes compensate each other’s expression maintaining protein native levelsPLOS ONE

Dear Dr. Cruz,

Thank you for submitting your manuscript to PLOS ONE. After careful consideration, we feel that it has merit but does not fully meet PLOS ONE’s publication criteria as it currently stands. Therefore, we invite you to submit a revised version of the manuscript that addresses the points raised during the review process.

We look forward to receiving your revised manuscript.

Kind regards,

Keisuke Hitachi

Academic Editor

PLOS ONE

Journal Requirements:

**Additional Editor Comments:**

As per the reviewer's comments, please correctly describe the scale. Also, please add the addition to Methods, as noted in the comments. The reviewer's comment about sequencing is valid, and I ask that you respond to it.

Reviewers' comments:

Reviewer's Responses to Questions

**Comments to the Author**

1. If the authors have adequately addressed your comments raised in a previous round of review and you feel that this manuscript is now acceptable for publication, you may indicate that here to bypass the “Comments to the Author” section, enter your conflict of interest statement in the “Confidential to Editor” section, and submit your "Accept" recommendation.

Reviewer #2: (No Response)

2. Is the manuscript technically sound, and do the data support the conclusions?

Reviewer #2: Partly

3. Has the statistical analysis been performed appropriately and rigorously? 

Reviewer #2: Yes

4. Have the authors made all data underlying the findings in their manuscript fully available?

Reviewer #2: Yes

5. Is the manuscript presented in an intelligible fashion and written in standard English?

Reviewer #2: (No Response)

6. Review Comments to the Author

Reviewer #2: Critique of PONE-D-23-29797_R3.

Reviewer #2: Thank you for your responses to R2. However, I see two outstanding issues:

1. Please define the calculation and scale for "band skew”. I brought this is in my comments on version R1.

We apologize for the laconic explanation given on R1. The band intensity on western blots was determined by ImageJ using “ROI manager” (ROI = Regions of Interest); the tool allows quantification of all bands, based on their size and intensity. Inside the ImageJ software, the array can be segmented into ROIs of identical size, where each ROI corresponds to one protein, certifying that the same area was considered for future comparisons between the samples. Thus, band skew shows how much the band intensity decreased comparing the RP expression in the presence or absence of its paralogue (Nat Protoc. 2016 August; 11(8): 1508–1530. doi:10.1038/nprot.2016.089).

Reviewer’s reply and comment to the authors answers to R3 comments

I know the standard procedure. However, "Band Skew" is, to my knowledge, not a standard term (I did not know it and I have read a lot of papers and worked in the field for a long time). As such Band Skew should be defined in Materials and Methods. And the scale should be defined (log2 or log10 change of expression). Note that I twice specifically requested that the scale be defined.

2. The myc insertions have not been sequenced. I think this should be done.

We will ask reviewer #2 to reconsider this specific demand for the reasons below. We agree that sequencing knockouts, tagging or any genome editing is the most robust form of confirming the expected modification, without any further undesirable event, or off-target insertion. Nevertheless, I want to raise some points that might convince Reviewer #2 that, in this case specifically, we have robust evidence that both knockout and myc tags occurred as aimed. First aspect to consider is that genome editing in Leishmania has been first achieved in 1990, using the classical method of gene replacement that had been previously employed for Saccharomyces cerevisiae. We (Cruz and Beverley, 1990) pioneered the field of Leishmania genetic manipulation when we showed that “Following introduction of a construct containing dihydrofolate reductase–thymidylate synthase (dhfr-ts) flanking sequences fused to neomycin phosphotransferase, 45% of the colonies contained the planned homologous replacement; this frequency rose to nearly 100% in transfections using low amounts of DNA. Integrative transfection in Leishmania thus resembles that of Saccharomyces cerevisae in giving predominantly homologous events.”

(http://dx.doi.org/10.1038/348171a0). This pioneer work demonstrated that the parasite has a highly efficient machinery of homologous recombination, which largely differs from most eukaryotes; insertion does not occur randomly and even when we exceeded in the amount of DNA to be transfected, what we observed was multiple insertions of the fragment in the homologous sequence/aimed site (same article above). The establishment of the CRISPR-Cas9 in these parasites (doi: 10.1098/rsos.170095) came to facilitate the previously labor-intensive and time-consuming approach, and represents a significant advance in the field. Even after 6 years of CRISPR/Cas9 use in Leishmania, we find no publications reporting off-target editing in Leishmania, and there are authors that report complete sequencing of the genome of transfected parasite, with no off-target events, this is probably partially due to the intrinsic genetic features of the organism. In addition to the abovementioned, we must consider that the results we report now compose a robust indication that myc insertion occurred in the expected genomic site.

The tag insertion in the RP genes studied in our work was also confirmed based on immune detection, as we used S16 antibody; the subcellular distribution of the tagged proteins using α-myc antibody was undistinguishable from the distribution of RPS16 protein detected by a specific α-S16 antibody (see supplementary material SF2). We have also used S16 polyclonal antibody in WB and the pattern was the same (not shown). The compensatory effect that maintains protein levels at similar levels comparing KO with wild type parasites was coherent, for all the paralogues (S16 and L13), showing that we are dealing with the protein we aimed to tag. For these reasons, we believe that the conventional PCR was an adequate tool in this study and confirmed the high efficiency of homologous recombination in Leishmania parasites, for both knockout and tagged parasites generation.

Reviewer reply

I am fully convinced of the authors’ experience in manipulating the Leishmania genome. However, my concern is not whether insertions and tags were successfully placed in the S16 genes. Rather, in spite of the efficiency of CRISPR, it seems possible that the target sequence is not copied accurately into the genome, which could lead to a change of the amino acid sequence of the tag, and thus a change in the antibody affinity for the tag. This should not affect the relative expression of a specific paralog but may affect the estimate of the relative expression of the two paralogs. Can the authors conclusively refute this scenario?

7. PLOS authors have the option to publish the peer review history of their article (what does this mean?). If published, this will include your full peer review and any attached files.

Reviewer #2: **Yes: **Lasse Lindahl

---

## [Author Response · Author response to Decision Letter 4]

31 Mar 2024

Critique of PONE-D-23-29797_R4.

Reviewer #2: I know the standard procedure. However, "Band Skew" is, to my knowledge, not a standard term (I did not know it and I have read a lot of papers and worked in the field for a long time). As such Band Skew should be defined in Materials and Methods. And the scale should be defined (log2 or log10 change of expression). Note that I twice specifically requested that the scale be defined.

Author’s comment: In image processing, skewness (or band skew) refers to a measure of the asymmetry of the pixel intensity distribution within an image (doi:10.1167/10.9.3). This parameter helps to quantify the degree to which the pixel values deviate from a symmetrical distribution around the mean. Thus, positive skewed distribution means that the right tail of the intensity values is longer or fatter than the left tail, indicating that there are more high-intensity pixels (doi:10.1167/10.9.3). So, in our case, we used the ROI manager toll from ImageJ software to determine the intensity of the bands, being directly correlated to the protein amount. Considering this, our comparisons present a positive increment in the band skew, showing an increment in the pixel’s intensity/protein amount. Additionally, skewness itself does not have a specific unit in the context of image processing. It is a statistical measure used to describe the shape of the pixel intensity distribution, providing information about the asymmetry of the distribution without being tied to a particular unit (https://brownmath.com/stat/shape.htm). We added such specification about skewness/band skew in the Methods section.

Reviewer #2: I am fully convinced of the authors’ experience in manipulating the Leishmania genome. However, my concern is not whether insertions and tags were successfully placed in the S16 genes. Rather, in spite of the efficiency of CRISPR, it seems possible that the target sequence is not copied accurately into the genome, which could lead to a change of the amino acid sequence of the tag, and thus a change in the antibody affinity for the tag. This should not affect the relative expression of a specific paralog but may affect the estimate of the relative expression of the two paralogs. Can the authors conclusively refute this scenario?

Author’s comment: All tags were included at the 5’-end of the genes. So, specific primers were used to amplify the tagged region, from the tag (myc) insertion to the CDS of each gene. Amplicons were cloned into pCR4-TOPO plasmid and sequenced by Sanger method using M13 primes. The results show that all RPs target genes investigated were successfully tagged using myc and with a high identity for the nucleotides, when aligned to the myc sequence. These results prove that the CRISPR/Cas9 system has efficiently drove the tag insertion at the target locus with accuracy of this sequence, avoiding changes in the final amino acid sequence. All sequencing results were added as supplementary material (SF.6) and allowed us to conclude that no misunderstanding in the relative expression of the paralogs has been done in the paper due to any divergence in the tag sequence.

---

## [Decision Letter · Decision Letter 5]

23 Apr 2024

Leishmania Ribosomal Protein (RP) paralogous genes compensate each other’s expression maintaining protein native levels

PONE-D-23-29797R5

Dear Dr. Cruz,

We’re pleased to inform you that your manuscript has been judged scientifically suitable for publication and will be formally accepted for publication once it meets all outstanding technical requirements.

Kind regards,

Alexander F. Palazzo, Ph.D.

Academic Editor

PLOS ONE

Additional Editor Comments (optional):

Reviewers' comments:

Reviewer's Responses to Questions

**Comments to the Author**

1. If the authors have adequately addressed your comments raised in a previous round of review and you feel that this manuscript is now acceptable for publication, you may indicate that here to bypass the “Comments to the Author” section, enter your conflict of interest statement in the “Confidential to Editor” section, and submit your "Accept" recommendation.

Reviewer #2: All comments have been addressed

2. Is the manuscript technically sound, and do the data support the conclusions?

Reviewer #2: Yes

3. Has the statistical analysis been performed appropriately and rigorously? 

Reviewer #2: Yes

4. Have the authors made all data underlying the findings in their manuscript fully available?

Reviewer #2: Yes

5. Is the manuscript presented in an intelligible fashion and written in standard English?

Reviewer #2: Yes

6. Review Comments to the Author

Reviewer #2: Thank you for explaining band skew and the accuracy of the myc tag in your revision. I accept your response to my comments.

7. PLOS authors have the option to publish the peer review history of their article (what does this mean?). If published, this will include your full peer review and any attached files.

Reviewer #2: **Yes: **Lasse Lindahl

---

## [Editor Report · Acceptance letter]

7 May 2024

PONE-D-23-29797R5 

PLOS ONE

Dear Dr. Cruz, 

I'm pleased to inform you that your manuscript has been deemed suitable for publication in PLOS ONE. Congratulations! Your manuscript is now being handed over to our production team.

Kind regards, 

on behalf of

Dr. Alexander F. Palazzo 

Academic Editor

PLOS ONE